# Scaling Up from Leaf to Whole-Plant Level for Water Use Efficiency Estimates Based on Stomatal and Mesophyll Behaviour in *Platycladus orientalis*

Yonge Zhang [1], Bing Liu [1], Guodong Jia [2,*], Xinxiao Yu [2], Xiaoming Zhang [1], Xiaolin Yin [1], Yang Zhao [1], Zhaoyan Wang [1], Chen Cheng [1], Yousheng Wang [1] and Yan Xin [1]

[1] State Key Laboratory of Simulation and Regulation of Water Cycle in River Basin, China Institute of Water Resources and Hydropower Research, Beijing 100038, China; zye0516@126.com (Y.Z.); liubingiwhr@163.com (B.L.); zxmwq@126.com (X.Z.); yxl0321@163.com (X.Y.); zhaoyang1224@163.com (Y.Z.); wangzhaoys@163.com (Z.W.); chengc@iwhr.com (C.C.); wangyousheng119@163.com (Y.W.); xinyan@iwhr.com (Y.X.)

[2] Key Laboratory of State Forestry Administration on Soil and Water Conservation, Beijing Forestry University, Beijing 100083, China; yuxinxiao1111@126.com

\* Correspondence: jgd3@163.com

**Abstract:** Prediction of whole-plant short-term water use efficiency ($WUE_{s,P}$) is essential to indicate plant performance and facilitate comparison across different temporal and spatial scales. In this study, an isotope model was scaled up from the leaf to the whole-plant level, in order to simulate the variation in $WUE_{s,P}$ in response to different $CO_2$ concentrations ($C_a$; 400, 600, and 800 $\mu mol \cdot mol^{-1}$) and soil water content (SWC; 35–100% of field capacity). For $WUE_{s,P}$ modelling, leaf gas exchange information, plant respiration, and "unproductive" water loss were taken into account. Specifically, in shaping the expression of the $WUE_{s,P}$, we emphasized the role of both stomatal ($g_{sw}$) and mesophyll conductance ($g_m$). Simulations were compared with the measured results to check the model's applicability. The verification showed that estimates of $g_{sw}$ from the coupled photosynthesis ($P_{n,L}$)-$g_{sw}$ model accounting for the effect of soil water stress slightly outperformed the model neglecting the soil water status effect. The established coupled $P_{n,L}$-$g_m$ model also proved more effective in estimating $g_m$ than the previously proposed model. Introducing the two diffusion control functions into the whole-plant model, the developed model for $WUE_{s,P}$ effectively captured its response pattern to different $C_a$ and SWC conditions. Overall, this study confirmed that the accurate estimation of $WUE_{s,P}$ requires an improved predictive accuracy of $g_{sw}$ and $g_m$. These results have important implications for predicting how plants respond to climate change.

**Keywords:** mesophyll conductance; stomatal conductance; stable isotope; soil water stress; water use efficiency; whole-plant level

---

## 1. Introduction

Water use efficiency (WUE), which refers to the ratio of carbon assimilation to water transpired by plants (i.e., water loss), is essential in optimizing plant water use [1]. The WUE can be defined at different temporal and spatial scales. At the leaf level, WUE describes the leaf net photosynthetic rate ($P_{n,L}$) relative to the leaf transpiration rate ($E_L$). Both processes are controlled by stomatal conductance ($g_{sw}$). The $P_{n,L}$ is also controlled by mesophyll conductance ($g_m$), in addition to $g_{sw}$, as recent studies demonstrated that mesophyll resistance is not negligible [2,3] and may be as important as stomatal conductance [4]. At the whole-plant level, all photosynthetic and non-photosynthetic parts contribute to respiration and water loss. However, the canopy accounts for the most significant part of carbon assimilation and transpiration water loss. Therefore, changes in $g_{sw}$ (and or $g_m$) may decrease or increase the whole-plant WUE, especially at smaller temporal scales.

Investigating whole-plant WUE at smaller temporal scales (hours or days) not only facilitates our understanding of whole-plant long-term (months, years, or decades) WUE and the underlying mechanism but also allows us to compare across different temporal and spatial scales. There have, however, been a few attempts to relate $g_{sw}$ (and or $g_m$) to whole-plant WUE at smaller temporal scales, or models to predict the response pattern of whole-plant short-term WUE ($WUE_{s,P}$) to environmental changes. The estimation of $WUE_{s,P}$ is frequently conducted on the assumption that leaf short-term WUE ($WUE_{s,L}$) is representative of $WUE_{s,P}$ [5]. However, there may be a gap between the daily integrals of leaf and whole-plant WUE, as carbon and water loss from non-photosynthetic tissue can result in a decrease in $WUE_{s,P}$ while not affecting $WUE_{s,L.}$ Therefore, it is critical to obtain adequate predictions of whole-plant WUE at smaller temporal scales.

It has been suggested that the leaf WUE model can be scaled to the whole-plant level by taking into account "unproductive" water loss and carbon use by respiration, independent of photosynthesis [6–8]. Built upon this concept, the Farquhar et al. (1989) [7] model relates leaf gas exchange properties and carbon discrimination to whole-plant WUE, but it ignores the effect of mesophyll resistance (the inverse of $g_m$) on carbon discrimination ($\Delta$). Thus, the contribution of $g_m$ to $\Delta$ needs to be considered [9], and that $g_m$ should have been incorporated in the approach of Farquhar et al. (1989) to predict whole-plant WUE accurately. This hypothesis was supported by our previous findings [10], which found that the whole-plant model emphasizing the role of $g_m$ outperformed the Farquhar et al. (1989) [7] model. Despite years of research, the three most widely used approaches for determining $g_m$, including the high number of gas exchange properties or measurements of gas exchange combined with chlorophyll fluorescence or carbon isotope discrimination [11], use complex parameters associated with complicated measurements, limiting the easy determination of $g_m$. In contrast, the soil water content and potential $g_m$ (unstressed $g_m$, $g_{m,p}$)-dependent empirical model proposed by Keenan et al. (2010) [12], can easily be used. Unfortunately, the model is still flawed in reflecting the influence of other environmental factors and gas exchange properties on $g_m$. A practical and relatively simple representation of mesophyll behaviour may lie at the heart of a valid and useful prediction of $WUE_{s,P}$. Furthermore, the revised whole-plant model [10] for $WUE_{s,P}$ included the presence of $g_{sw}$, in addition to $g_m$, thereby representing the linkage between $WUE_{s,P}$ and $g_{sw}$. Although several models have been proposed to describe stomatal behaviour, including the simple coupled photosynthesis–stomatal conductance ($P_{n,L}$-$g_{sw}$) model and its modified versions, it remains unclear which approach is the most useful. In general, the WUE model scaling from the leaf to the whole-plant level needs to be revised and improved based on well-modelled stomatal and mesophyll behaviors.

The latest observations showed that globally-averaged atmosphere $CO_2$ concentration ($C_a$) reached a new high ($413.2 \pm 0.2$ µmol·mol$^{-1}$) in 2020 [13]. If the upward trend of $C_a$ continues, soil water stress may be intensified by climate change in many areas. Making it crucial to predict how $WUE_{s,P}$ responds to the different $C_a$ and soil water content (SWC). Therefore, we developed a model to estimate $g_m$ based on the empirical relationship between $g_m$ and $P_{n,L}$ (i.e., the coupled photosynthesis-mesophyll conductance model), and the revised $g_m$ model and the previously established $g_{sw}$ model were then incorporated into the whole-plant WUE model to estimate the variation in $WUE_{s,P}$. Measurements of whole-plant net $CO_2$ gas exchange (root systems have been excluded from measurements, i.e., aboveground measurements) and transpiration under different $C_a \times$ SWC conditions were conducted concurrently, allowing us to calculate the actual $WUE_{s,P}$ and to compare the measured results with simulations obtained from the developed whole-plant WUE model. Our aim was, first, to establish a reliable model for $g_m$; second, to check the applicability of the whole-plant WUE model scaled from the leaf level, based on estimations of stomatal and mesophyll behavior.

## 2. Theoretical Background

### 2.1. Coupled $g_{sw}$-$P_{n,L}$ Model

Previous studies found that leaf stomatal conductance ($g_{sw}$, mol $H_2O \cdot m^{-2} \cdot s^{-1}$) is highly correlated with photosynthesis ($P_{n,L}$, $\mu mol \cdot m^{-2} \cdot s^{-1}$). Based on this, a series of models on the basis of the linear relationship between $g_{sw}$ and $P_{n,L}$ have been proposed [14–16]. By incorporating the effect of leaf-to-air vapor pressure deficit (D), Leuning et al. (1995) [17] established an alternative coupled $P_{n,L}$-$g_{sw}$ model based on former studies

$$g_{sw} = g_{0,sw} + g_1 P_{n,L} \frac{f(D)}{C_s - \Gamma}$$ (1)

where $g_{0,sw}$ and $g_1$ are fitted parameters and $g_{0,sw}$ is considered to represent the residual stomatal conductance (mol $H_2O \cdot m^{-2} \cdot s^{-1}$); $C_s$ is the leaf surface $CO_2$ concentration ($\mu mol \cdot mol^{-1}$); $\Gamma$ is the $CO_2$ compensation point ($\mu mol \cdot mol^{-1}$); $f(D)$ is the vapor pressure deficit-dependent function. To describe the effect of $D$ on stomatal behaviour, numerous expressions have been introduced [14,17–21]. Lloyd (1991) [19] and Yu et al. (2001) [21] consistently found that the precision of estimation was highest when imposing the function $f(D) = h_s$, with $h_s$ referring to the relative humidity at leaf surface in %. Thus, we adopted the expression $f(D) = h_s$ in the Leuning et al. (1995) model [17].

The model introduced by Leuning et al. (1995) [17] has been widely used to predict gas exchange properties at the leaf scale [16,22], albeit without taking into account the response of water stress. To overcome this limitation, Egea et al. (2011) [23] proposed an improved model, which incorporated a soil water stress-dependent function ($f(\theta_s)$, calculated by Equation (5)), to describe the behaviour of gas exchange properties

$$g_{sw} = g_{0,sw} + g_1 P_{n,L} \frac{f(\theta_s)f(D)}{C_s - \Gamma}$$ (2)

### 2.2. Coupled $g_m$-$P_n$ Model

Models which can easily represent mesophyll behaviour in response to environmental drivers are still scarce. Considering the restrictions of soil water stress on $g_m$ (mol $CO_2 \cdot m^{-2} \cdot s^{-1}$), Keenan et al. (2010) [12] proposed a function to predict the linkage between $g_m$ and soil water status

$$g_m = f(\theta_m)g_{m,p}$$ (3)

where $f(\theta_m)$ is the mesophyll conductance limitation function, which depends on soil water stress (calculated by Equation (5)); $g_{m,p}$ is the potential (unstressed) $g_m$. This model has been used to represent the feedback of $g_m$ to soil water stress [1], but does not consider the response of mesophyll behaviour to other environmental drivers, such as $C_a$. In fact, $g_m$ is affected by increases or decreases in $C_a$, and even changes more subtly with changes in $C_a$ than in $g_{sc}$ ($g_{sc} = g_{sw}/1.6$) [24]. Previous studies have observed that the $P_{n,L}$ increased linearly with $g_m$ [25–27], which prompted us to establish a coupled $P_{n,L}$-$g_m$ function to model $g_m$ by imposing similar limitation functions to mesophyll behavior as those imposed to stomatal behaviour. Based on the empirical relationship between $P_{n,L}$ and $g_m$, the proposed model is as follows

$$g_m = g_{0,m} + g_2 P_{n,L} \frac{f(\theta_m)f(D)}{C_s - \Gamma}$$ (4)

where $g_{0,m}$ and $g_2$ are fitted parameters, and $g_{0,m}$ is considered to represent the residual mesophyll conductance (mol $CO_2 \cdot m^{-2} \cdot s^{-1}$).

The two soil water stress-dependent limitation functions, $f(\theta_s)$ and $f(\theta_s)$, were expressed as [12,28]

$$f(\theta_i) = \begin{cases} 1 & \theta \geq \theta_c \\ \left[\frac{\theta - \theta_w}{\theta_c - \theta_w}\right]^{q_i} & \theta_w \leq \theta \leq \theta_c \\ 0 & \theta \leq \theta_w \end{cases} \tag{5}$$

where $\theta$ is the soil volumetric water content (%); $\theta_c$ and $\theta_w$ are soil water content levels at field capacity (26.20%) and permanent wilting point (4.08%), respectively; parameter $q_j$ is a measure of the nonlinearity of the effects of soil water stress on the limiting mechanisms; the subscript i = s and m represent stomatal and mesophyll limitations, respectively. In this study, the selected values for tunable parameters of $q_s$ and $q_m$ were 0.25, 0.50, 0.75, 1.00, 1.25, and 1.50, within the previously reported range [12,23].

### 2.3. Leaf and Whole-Plant WUE Model

The leaf instantaneous water use efficiency ($WUE_{i,L}$, mmol·mol$^{-1}$) is the ratio of leaf net photosynthetic rate ($P_{n,L}$, μmol·m$^{-2}$·s$^{-1}$) to transpiration rate ($E_L$, mmol·m$^{-2}$·s$^{-1}$) [6]

$$WUE_{i,L} = \frac{P_{n,L}}{E_L} = \frac{P_{n,L}}{g_{sw}D} \tag{6}$$

Substituting the Egea et al. (2011) [23] model (Equation (2)) and the Leuning et al. (1995) [17] model (Equation (1)) into Equation (6), we obtain the following formulas, respectively

$$WUE_{i,L} = \frac{P_{n,L}}{D} \times \frac{C_s - \Gamma}{(C_s - \Gamma)g_{0,sw} + g_1 P_{n,L} f(\theta_s) f(D)} \tag{7}$$

$$WUE_{i,L} = \frac{P_{n,L}}{D} \times \frac{C_s - \Gamma}{(C_s - \Gamma)g_{0,sw} + g_1 P_{n,L} f(D)} \tag{8}$$

The $WUE_{i,L}$ inferred from Equation (7) with well parameterized $q_s$ ($q_s$ = 0.25, see Section 4.1) is model configuration 1, and that inferred from Equation (8) is model configuration 2.

The whole-plant instantaneous water use efficiency ($WUE_{i,P}$, mmol·mol$^{-1}$) is the ratio of whole-plant net photosynthetic rate ($P_{n,p}$, μmol·h$^{-1}$) to transpiration rate ($E_p$, mmol·h$^{-1}$) [7]. Considering respiration and water loss from the non-photosynthetic organs, the ratio of instantaneous net photosynthesis to transpiration can be scaled from the leaf to the whole-plant level

$$WUE_{i,P} = \frac{P_{n,P}}{E_p} = \frac{P_{n,L}}{E_L} \times \frac{(1 - \phi_{c,i})}{(1 + \phi_{w,i})} = \frac{P_{n,L}}{g_{sw}D} \times \frac{(1 - \phi_{c,i})}{(1 + \phi_{w,i})} \tag{9}$$

where $\phi_{c,i} = (3.6 P_{n,L} \times LA - P_{n,P})/(3.6 P_{n,L} \times LA)$, with $LA$ referring to plant total leaf area in m$^2$) is the proportion of respiration from non-photosynthetic parts (twigs and stem) during the daytime, and $\phi_{w,i} = (E_P - 3.6 E_L \times LA)/(3.6 E_L \times LA)$ is the proportion of water loss from non-photosynthetic parts during the daytime. Similarly, we substituted the simulated $g_{sw}$, calculated via the Egea et al. (2011) [23] model (Equation (2)) and the Leuning et al. (1995) [17] model (Equation (1)) into Equation (9), obtaining the following formulas, respectively

$$WUE_{i,P} = \frac{P_{n,L}}{D} \times \frac{C_s - \Gamma}{(C_s - \Gamma)g_{0,sw} + g_1 P_{n,L} f(\theta_s) f(D)} \times \frac{(1 - \phi_{c,i})}{(1 + \phi_{w,i})} \tag{10}$$

$$WUE_{i,P} = \frac{P_{n,L}}{D} \times \frac{C_s - \Gamma}{(C_s - \Gamma)g_{0,sw} + g_1 P_{n,L} f(D)} \times \frac{(1 - \phi_{c,i})}{(1 + \phi_{w,i})} \tag{11}$$

The whole-plant short-term water use efficiency ($WUE_{s,P}$) is the ratio of whole-plant cumulative $CO_2$ assimilation to water loss. At the diel time scale, not only the role of

respiration and water loss from non-photosynthetic parts (twigs and stem) during the daytime need to be included, but also respiration and water loss from whole parts (leaf, twigs, and stem) during the nighttime contribute substantially to $\text{WUE}_{s,P}$. When all these processes are taken into account, the time-integrated $\text{WUE}_{s,P}$ is as follows

$$\text{WUE}_{s,P} = \frac{\int P_{n,P}}{\int E_P} = \frac{\int P_{n,L}}{\int E_L} \times \frac{(1 - \phi_{c,s})}{(1 + \phi_{w,s})} = \frac{\int P_{n,L}}{\overline{g_{sw} \overline{D}}} \times \frac{(1 - \phi_{c,s})}{(1 + \phi_{w,s})} \tag{12}$$

where $\phi_{c,s} = (3.6\, P_{n,L} \times LA - P_{n,P} + R_P)/(3.6\, P_{n,L} \times LA)$, with $R_P$ referring to night-time respiration in $\text{mmol·h}^{-1}$) is the proportion of respiration from non-photosynthetic parts (twigs and stem) during the whole time and from leaves during the nighttime; $\phi_{w,s} = (E_P - 3.6\, E_L \times LA + E_d)/(3.6\, E_L \times LA)$, with $E_d$ referring to nighttime transpiration in $\text{mol·h}^{-1}$) is the proportion of water loss from non-photosynthetic parts (twigs and stem) during the whole time and from leaves during the nighttime. The above time integral is denoted as $\int$. According to Fick's law

$$\frac{P_{n,L}}{g_{sw}} = \frac{C_a}{1.6} \times (1 - \frac{C_i}{C_a}) \tag{13}$$

where $C_i$ is the leaf intercellular $CO_2$ concentration ($\mu\text{mol·mol}^{-1}$). The photosynthetic $^{13}C$ discrimination ($\Delta$, ‰) reflects the physiological properties over short time scales [29–31]. From the variant of the Farquhar et al. (1989) [7] classical model, including the effect of $g_m$ on $\Delta$, the short-term $C_i/C_a$ ratio can be written as follows

$$\frac{C_i}{C_a} = \frac{\Delta_{mea} - a + (b - a_m)\frac{g_{sw}}{1.6 g_m}}{b - a + (b - a_m)\frac{g_{sw}}{1.6 g_m}} \tag{14}$$

where $a$ is the fractionation associated with the atmospheric $CO_2$ diffusion at the boundary layer (4.4‰); $a_m$ is the fractionation of $CO_2$ diffusion and dissolution in the liquid phase (1.8‰); $b$ is the fractionation during carboxylation (29‰); $\Delta_{mea}$ is measured photosynthetic $^{13}C$ discrimination $= (\delta^{13}C_a - \delta^{13}C_l)/(1 + \delta^{13}C_l)$, with $\delta^{13}C_a$ and $\delta^{13}C_l$ referring to $\delta^{13}C$ of atmospheric $CO_2$ and water-soluble organic materials (WSOM, fast-turn-over carbohydrates) in leaves, respectively.

Substituting Equation (13) and Equation (14) into Equation (12), we obtain the following equation

$$\text{WUE}_{s,P} = \frac{C_a}{1.6D} \times \frac{b - \Delta}{b - a + (b - a_m)\frac{g_{sw}}{g_m}} \times \frac{(1 - \phi_{c,s})}{(1 + \phi_{w,s})} \tag{15}$$

Similar to the simulation of $\text{WUE}_{i,L}$, two model configurations were applied in Equation (15), and we obtained the following equations

$$\text{WUE}_{s,P} = \frac{C_a}{1.6D} \times \frac{(1 - \phi_{c,s})}{(1 + \phi_{w,s})} \times \frac{b - \Delta}{b - a + (b - a_m) \times \frac{(C_s - \Gamma)g_{0,sw} + g_1 P_{n,L} f(\theta_s) f(D)}{(C_s - \Gamma)g_{0,m} + g_2 P_{n,L} f(\theta_m) f(D)}} \tag{16}$$

$$\text{WUE}_{s,P} = \frac{C_a}{1.6D} \times \frac{(1 - \phi_{c,s})}{(1 + \phi_{w,s})} \times \frac{b - \Delta}{b - a + (b - a_m) \times \frac{(C_s - \Gamma)g_{0,sw} + g_1 P_{n,L} f(\theta_s) f(D)}{(C_s - \Gamma)f(\theta_m)g_{m,p}}} \tag{17}$$

## 3. Material and Methods

### 3.1. Experimental Design and Management

The experiment was carried out in April 2018 at the Chinese Forest Ecosystems Research Network (116°05′ E, 40°03′ N), situated at the Western Hill, Beijing, North China, using 7-year-old *Platycladus orientalis* saplings of the same genotype of a temperate origin. The plants were each transplanted into 15.51-L pots containing soil collected from a local *Platycladus orientalis* stand. The soil type is sandy loam, and the field capacity ($\theta_c$, 26.2%)

and permanent wilting point ($\theta_w$, 4.08%) of the soil and plants were determined by a pilot experiment. The $\theta_c$ was measured by soil water content (SWC) sensors (HOBO–U30, Onset, Cape Cod, Massachusetts, USA) after soil samples absorbed water for 24 h with no vertical underwater droplets. The $\theta_w$ was determined by the same sensors when leaves produced wilting and could not be restored by supplemental water, that is, below the wilting point leaf water potential (measured by portable plant water potential meter (WP4C, Decagon, Pullman, WA, USA); data not shown) did not increase with the increase in SWC. *Platycladus orientalis* samplings with similar growth status and canopy structure (approximately 1.4 m high) were grown in a greenhouse. After acclimation in the greenhouse for two months, saplings were moved to growth chambers (FH-230, Taiwan Hipoint Corporation, Kaohsiung City, Taiwan) and subjected to a nested design with three $CO_2$ concentration ($C_a$) levels and five SWC regimes. The controlled environment (light, air temperature, and relative humidity) in the growth chambers was set to simulate natural growth conditions. From 07:00 to 19:00 (simulating daytime), all white LED lights were turned on, with 60% relative humidity and 25 °C. From 19:00 to 07:00 (simulating nighttime), all white LED lights were turned off, with 80% relative humidity and 18 °C. In North China, *P. orientalis* saplings are generally grown under the forest canopy, which receives a lower photosynthetic photon flux density (with an average of $230 \pm 37$ µmol·m$^{-2}$·s$^{-1}$) than full sunlight (with an average of $350 \pm 41$ µmol·m$^{-2}$·s$^{-1}$) at daytime during the growing season. Thus, the low level of light intensity in the growth chamber ($220 \pm 20$ µmol·m$^{-2}$·s$^{-1}$) was considered to be approximately appropriate to simulate the growth of understory saplings.

To realize orthogonal treatments, two growth chambers were used. One growth chamber (Figure 1a) was connected to a $CO_2$ tank and ambient atmosphere with an intake pipe, which was used to maintain elevated $C_a$ of 600 µmol·mol$^{-1}$ ($C_{600}$) or 800 µmol·mol$^{-1}$ ($C_{800}$). Another growth chamber (Figure 1b) was only connected to ambient atmosphere with an intake pipe to maintain $C_a$ of approximately 400 µmol·mol$^{-1}$ ($C_{400}$). $CO_2$ sensors and control systems inside the growth chambers can continuously monitor and adjust $C_a$ steady near the enactment value, with a standard deviation of 50 µmol·mol$^{-1}$. Each $C_a$ treatment was subjected to five SWC regimes: (1) 35–45% of field capacity, FC, (simulating severe drought), (2) 50–60% of FC (moderate drought), (3) 60–70% of FC (mild drought), (4) 70–80% of FC (well-watered), and (5) 95–100% of FC (excessively watered). The FC of the potting soil was 26.20%. For the sake of calculative simplicity, we assumed that the SWC gradient was: (1) 10.48%, (2) 14.41%, (3) 17.03%, (4) 19.65%, and (5) 26.20%, respectively. The SWC in the upper 10 to 15 cm was continuously measured by sensors (HOBO–U30, Onset, Cape Cod, Massachusetts, USA), and the water status of each potting soil was checked twice daily and irrigated manually to achieve the target SWC regimes. The surface of the potting soil was covered with an approximately 2-cm layer of perlite to reduce soil evaporation. Each treatment ($C_a \times$ SWC) lasted for 30 days and had three pot-grown saplings as replicates. As one growth chamber was able to hold five pots, the experiment was performed progressively from June to November 2018, where treatments were maintained at $C_{400} \times$ SWC (in chamber b) and $C_{600} \times$ SWC (in chamber a) from June to August, and at $C_{400} \times$ SWC (in chamber b) from September to November. The pots were rearranged frequently to exclude position effects.

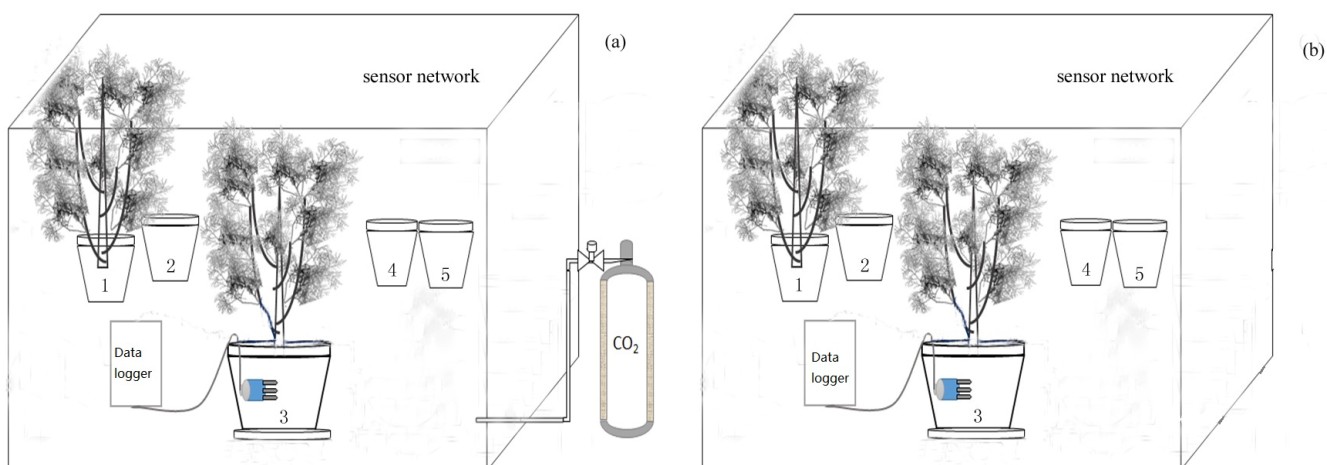

**Figure 1.** Schematic of growth chambers used in the experiments. One growth chamber (**a**) was used to maintain elevated $CO_2$ concentration of 600 µmol·mol$^{-1}$ or 800 µmol·mol$^{-1}$. Another growth chamber (**b**) was used to maintain $CO_2$ concentration of 400 µmol·mol$^{-1}$. There were five pots inside each chamber.

### 3.2. Measurements

#### 3.2.1. Whole-Plant Carbon Balance and Measurement

After the saplings had been subjected to the 30-day $C_a \times$ SWC treatment, whole-plant carbon balance was measured inside the growth chambers using the static chamber as designed by Jasoni et al. (2005) [32]. The static chamber measured $50 \times 50 \times 150$ cm, and in its interior, a pocket weather meter was incorporated (Kestrel 5500, Nielsen-Kellerman, Boothwyn, PA, USA) to monitor air temperature ($T_a$, K) and pressure ($P$, P$_a$). To avoid soil respiration, the substrate surface was tightly sealed with airtight plastic film as described by Escalona et al. (2013) [33]. Prior to each measurement, the sapling was enclosed in the static chamber, and the fan on its top turned on for 30 s to ensure that the flux was mixed well. The $C_a$ in the static chamber was measured by an infrared gas analyzer (Li-8100, Li-Cor, Lincoln, NE, USA), starting after the flux was well mixed (initial $C_a$, i.e., $C_0$) and finishing after the measurement had lasted for 3 min (final $C_a$, i.e., $C_1$). Measurements for each sapling were repeated three times and conducted at 9:00–10:00, 13:00–14:00, and 17:00–18:00 during daytime and at 22:00–23:00, 2:00–3:00, and 6:00–7:00 during nighttime. During the 3 min, the $C_a$ in the closed static chamber gradually decreased in the day but increased in darkness. The whole-plant daytime net photosynthetic rate ($P_{n,p}$) and the nighttime respiratory rate ($R_p$) were calculated as follows

$$P_{n,P} = \frac{V}{\Delta t} \times \frac{273.15}{T_a} \times \frac{P}{101,325} \times \frac{1}{22.41} \times (C_0 - C_1) \times \frac{60}{1000} \tag{18}$$

$$R_{n,P} = \frac{V}{\Delta t} \times \frac{273.15}{T_a} \times \frac{P}{101,325} \times \frac{1}{22.41} \times (C_1 - C_0) \times \frac{60}{1000} \tag{19}$$

where $V$ is the chamber volume (L) and $\Delta t = 3$ min is the time duration. The $P_{n,p}$ and R$_p$ were calculated from values measured during daytime and nighttime, respectively.

#### 3.2.2. Whole-Plant Transpiration Measurements

The whole-plant daytime transpiration rate ($E_p$) was measured from the beginning until the end of the experiment by a Flow 32-1K system (Dynamax, Houston, TX, USA). The Flow 32-1K system includes gauges installed at approximately 25 cm above the stem base and a CR1000 logger (Campbell Scientific, Logan, UT, USA), which continuously collected $E_p$ data every 15 min. In this study, each treatment ($C_a \times$ SWC) lasted for 30 days. The $E_p$ values remained relatively stable from the 21st day of orthogonal treatments, which were used for data analysis.

The whole-plant nighttime transpiration rate ($E_d$) was measured by mass loss during the night. Total plant nighttime transpiration was obtained from the difference in pots weight at the onset (19:00) and end of night (7:00). During plant nighttime transpiration measurements, the substrate surface was tightly sealed with airtight plastic film as described by Escalona et al. (2013) [33] to avoid soil evaporation. Measurements were made every 3 days.

The measured WUE$_{i,P}$ was the ratio between $P_{n,p}$ to $E_p$ ($P_{n,p}/E_p$), while the modelled WUE$_{i,P}$ was calculated by different model configurations. In model configuration 1 (Equation (10)), $g_{sw}$ was calculated by Equation (2) with well parameterized $q_s$ ($q_s = 0.25$, see Results 3.1). The model configuration 2 is Equation (11) with no additional parameterization associated with soil water stress.

The measured WUE$_{s,P}$ was the ratio between accumulative carbon gain and cumulative water loss, that is, WUE$_{s,P} = (P_{n,P} - R_P)/(E_P + E_d)$. In contrast, the modelled WUE$_{s,P}$ were calculated by different model configurations. In model configuration 1 (Equation (16)), $g_m$ was calculated by Equation (4) with well parameterized $q_m$ ($q_m = 0.25$, see Section 4.2), and $g_{sw}$ was calculated by Equation (2) with well parameterized $q_s$ ($q_s = 0.25$, see Section 4.1). In model configuration 2 (Equation (17)), $g_m$ was calculated by Equation (3) with well parameterized $q_m$ ($q_m = 0.50$, see Section 4.2), and $g_{sw}$ was calculated by Equation (1).

### 3.2.3. Leaf Gas Exchange and Stable Isotope Analysis

On the day of whole-plant carbon balance measurements, leaf gas change properties ($P_{n,L}$, $E_L$, $g_{sw}$, and $C_i$), leaf temperature ($T_L$), and leaf surface relative humidity ($RH$) were measured inside the growth chambers on mature leaves, using a portable gas exchange system (Li-6400, Li-Cor, Lincoln, NE, USA) fitted with a needle leaf chamber. The measurements were conducted at different positions (upper, middle, and lower crown) and made on at least three different leaves in each canopy layer at 9:00, 13:00, and 17:00. No significant differences ($p > 0.05$) in these measurements among different canopy layers were observed. Almost all leaves were exposed to similar light intensities and, thus, the effect of internal leaves was not considered. In this study, we assumed that a period of 30 days was long enough for saplings to be subjected to the treatments, according to our pilot experiment as described by Zhang et al. (2019) [10]. Measured leaf instantaneous water use efficiency (WUE$_{i,L}$) was calculated as the ratio between $P_{n,L}$ and $E_L$ ($P_{n,L}/E_L$).

The leaves used for gas exchange measurements were detached, immediately wrapped in tinfoil, and preserved in liquid nitrogen. Leaf water-soluble organic matter (WSOM) was extracted using the same method as described by Zhang et al. (2019) [10]. The obtained WSOM was dried and then combusted in an elemental analyzer (Flash EA 1112, Thermo Finnigan, California, USA) coupled to a continuous-flow stable isotope ratio mass spectrometer (DELTAplusXP, Thermo Finnigan, California, USA). The $\delta^{13}C$ of leaf WSOM ($\delta^{13}C_l$) was analyzed using the stable isotope ratio mass spectrometer with a precision of $\pm 0.1‰$. In addition, at the end of each treatment, atmosphere samples from the growth chamber were also collected (at least three replicates), and the $\delta^{13}C$ of the atmosphere ($\delta^{13}C_a$) was analyzed by the stable isotope ratio mass spectrometer. Measured $g_m$ was obtained by carbon isotope discrimination combined with gas exchange measurements as previously described by Zhang et al. (2019) [10], i.e.,

$$g_m = \frac{(b - a_i) \times \frac{P_{n,L}}{C_a}}{(\Delta_{lin} - \Delta_{mea})} \tag{20}$$

where $a_i$ is the fractionation of $CO_2$ diffusion and dissolution in the liquid phase (1.8‰), and $\Delta_{lin}$ is photosynthetic $^{13}C$ discrimination (‰) calculated by the version of the Farquhar et al. (1982) [34] simple linear model, namely,

$$\Delta_{lin} = a + (b' - a) \, C_i : C_a \tag{21}$$

where $b'$ is the fractionation relevant to the reactions of Rubisco and PEP carboxylase (27‰) [6].

### 3.2.4. Whole-Plant Total Leaf Area Measurement

At the end of the experiment, saplings were harvested and separated into different parts. A portion of leaves with different widths and shapes were selected as subsamples. Leaf subsample fresh weight ($FW_{sub}$) was immediately determined using electronic balance with an accuracy of $\pm$ 0.001 g, and the leaf area for subsample ($LA_{sub}$) was determined using image processing software for Photoshop. Subsequently, these leaves were dried at 80 °C for 48 h in an oven to obtain their dry weight ($DW_{sub}$). The dry weights of the remaining harvested leaves ($DW_{rest}$) were also determined. The whole-plant total leaf area (LA) of each sapling was calculated as follows

$$LA = R_D \times DW = (LA_{sub}/DW_{sub}) \times (DW_{sub} + DW_{rest} + DW_{iso}) \tag{22}$$

In this equation, $R_D$ is leaf area per dry weight ($m^2 \cdot g^{-1}$), $DW$ is whole-plant total dry weight (g), and $DW_{iso} = FW_{iso} \times DW_{sub}/FW_{sub}$, with $FW_{iso}$ referring to fresh weight of leaves used for isotope analysis in g) is dry weight of leaves used for isotope analysis (g).

### 3.3. Data Analysis

All statistical analyses were conducted using SPSS 19.0. The influences of $C_a$ and SWC on mean variables of $g_{sw}$, $g_m$, and WUE (including $WUE_{i,L}$, $WUE_{i,P}$, and $WUE_{s,P}$) were determined by two-way analysis of variance (ANOVA), and results were considered statistically significant at $p < 0.05$. Deviations of the modeled $g_{sw}$, $g_m$, and WUE from their measurements were absolute differences between the modeled and measured values. Relationships between the measured and modeled values in $g_{sw}$, $g_m$, and WUE were assessed using general linear regression analysis.

## 4. Results

### 4.1. Measured and Modelled Responses of $g_{sw}$ to SWC and $C_a$

Changes in SWC and $C_a$ significantly affected $g_{sw}$ ($p < 0.05$), with a maximum of 0.0963 mmol $H_2O \cdot m^{-2} \cdot s^{-1}$ at $C_{400} \times 19.65\%$ of SWC and a minimum of 0.0155 mol $H_2O \cdot m^{-2} \cdot s^{-1}$ at $C_{800} \times 10.48\%$ of SWC (Figure 2). In all cases, $g_{sw}$ decreased with elevated $C_a$. The $g_{sw}$ increased sharply as water stress was alleviated irrespective of $C_a$, and this effect was less evident when SWC exceeded 17.03% and even decreased when $g_{sw}$ peaked at 19.65% of SWC (Figure 2).

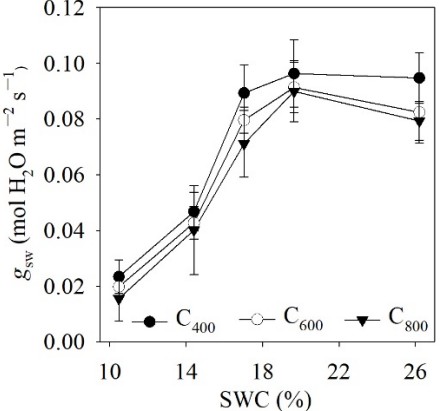

**Figure 2.** Response of measured leaf stomatal conductance ($g_{sw}$, mol $H_2O \cdot m^{-2} \cdot s^{-1}$) to three $CO_2$ concentrations ($C_a$) × five soil water contents (SWC). $C_{400}$, $C_{600}$, and $C_{800}$ are $C_a$ of 400, 600, and 800 μmol·mol$^{-1}$. Data represent mean values ± SD.

The $g_{sw}$ simulated by the two coupled $P_{n,L}$-$g_{sw}$ model (Equations (1) and (2)) decreased in response to elevated $C_a$ (Figure 3). In the absence of additional parameterization associated with soil water stress (Equation (1)), the $g_{sw}$ increased as the soil water status improved and reached maximum values at 19.65% of SWC, with a slight decrease thereafter. In contrast, when the effect of soil water stress was incorporated in the coupled $P_{n,L}$-$g_{sw}$ model (Equation (2)), the simulated $g_{sw}$ generally increased as SWC increased, regardless of the value imposed by $q_s$ (Figure 3).

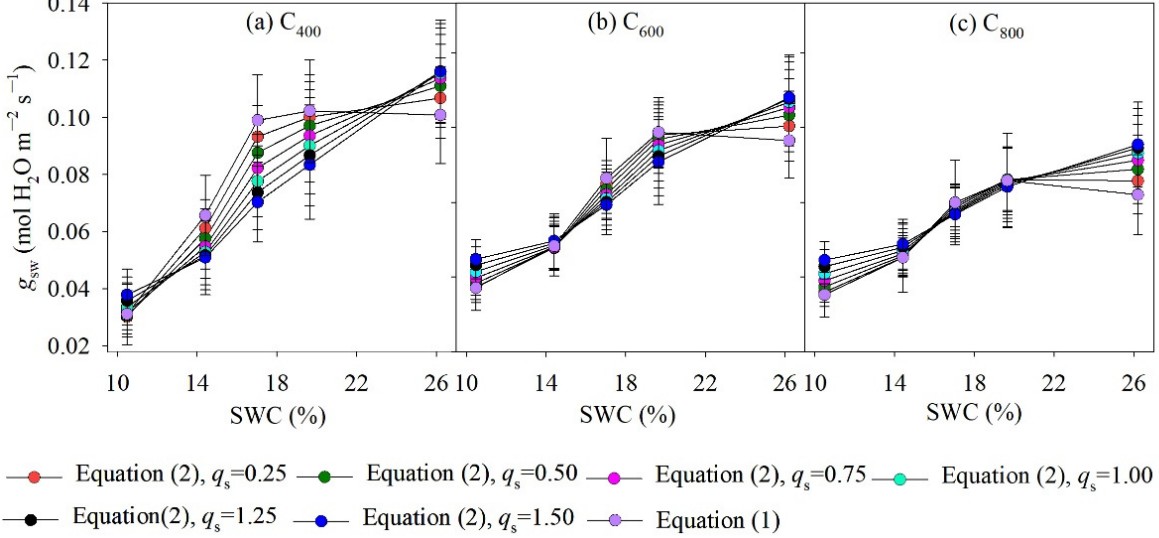

**Figure 3.** The estimated leaf stomatal conductance ($g_{sw}$, mol $H_2O\cdot m^{-2}\cdot s^{-1}$) in *Platycladus orientalis* saplings under different soil water contents (SWC) and $CO_2$ concentrations ($C_a$) conditions, based on different models (Equations (1) and (2)). $C_{400}$, $C_{600}$, and $C_{800}$ are $C_a$ of 400 (**a**), 600 (**b**), and 800 μmol·mol$^{-1}$ (**c**). Tunable parameter $q_s$ is a measure of the nonlinearity of the effects of soil water stress on the stomatal limiting mechanisms. Data represent mean values $\pm$ SD.

The correlation between the measured and calculated $g_{sw}$ is shown in Table 1. When applying Equation (2), we found a strong correlation between the calculated and the measured $g_{sw}$ ($p < 0.01$), and the correlation coefficient $R^2$ decreased from 0.88 to 0.68 as $q_s$ increased from 0.25 to 1.50. The calculated $g_{sw}$ based on Equation (1) also significantly correlated with the measured $g_{sw}$ ($p < 0.01$; $R^2 = 0.87$). However, when applying Equation (2), at $q_s = 0.25$, the calculated $g_{sw}$ (higher $R^2$ and slope closer to (1) was closer to measured $g_{sw}$ than when using Equation (1). Additionally, with Equation (2), there was less deviation ($0.0084 \pm 0.0053$ mol $H_2O\cdot m^{-2}\cdot s^{-1}$) between the measured and calculated $g_{sw}$ than with Equation (1) ($0.0086 \pm 0.0062$ mol $H_2O\cdot m^{-2}\cdot s^{-1}$). This showed that the $P_{n,L}$–$g_{sw}$ model, which incorporates the soil water stress ($q_s = 0.25$, Equation (2)), better predicts $g_{sw}$ than Equation (1).

**Table 1.** Correlation analysis between measured and modeled leaf stomatal conductance ($g_{sw}$, mol $H_2O\cdot m^{-2}\cdot s^{-1}$) using different models (Equations (1) and (2)).

| Model | Regression of Measured and Modelled Leaf $g_{sw}$ | | |
|---|---|---|---|
| | Linear Regression Equation | $R^2$ | $p$ |
| Equation (2), $q_s = 0.25$ | $y = 0.88x + 0.01$ | 0.88 | <0.01 |
| Equation (2), $q_s = 0.50$ | $y = 0.86x + 0.01$ | 0.86 | <0.01 |
| Equation (2), $q_s = 0.75$ | $y = 0.83x + 0.01$ | 0.83 | <0.01 |
| Equation (2), $q_s = 1.00$ | $y = 0.79x + 0.01$ | 0.79 | <0.01 |
| Equation (2), $q_s = 1.25$ | $y = 0.74x + 0.02$ | 0.74 | <0.01 |
| Equation (2), $q_s = 1.50$ | $y = 0.68x + 0.02$ | 0.68 | <0.01 |
| Equation (1) | $y = 0.87x + 0.01$ | 0.87 | <0.01 |

### 4.2. Measured and Modelled Responses of $g_m$ to SWC and $C_a$

The $g_m$ ranged between 0.0131 and 0.0571 mol $CO_2 \cdot m^{-2} \cdot s^{-1}$, significantly lower than $g_{sw}$ ($p < 0.05$). Elevation of $C_a$ produced significant changes in $g_m$. In all case, elevated $C_a$ decreased $g_m$ (Figure 4). Additionally, SWC significantly influenced the $g_m$ ($p < 0.05$) in a similar pattern as $g_{sw}$. Under low soil moisture content, $g_m$ increased rapidly with SWC. However, the rate of increase in $g_m$ decreased when SWC exceeded 17.03% and even decreased at SWC between 19.65% and 25.55% (Figure 4).

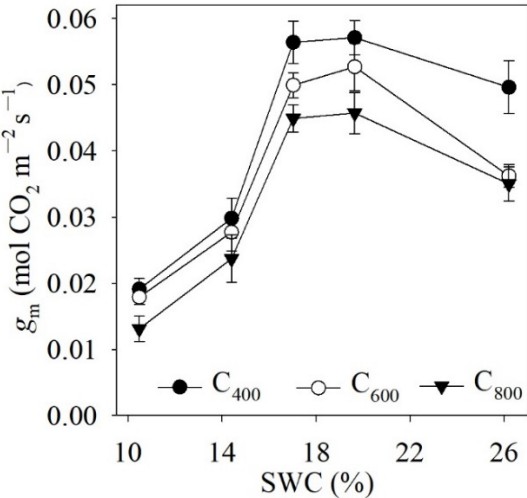

**Figure 4.** Response of measured leaf mesophyll conductance ($g_m$, mol $CO_2 \cdot m^{-2} \cdot s^{-1}$) to three $CO_2$ concentrations ($C_a$) × five soil water contents (SWC). $C_{400}$, $C_{600}$, and $C_{800}$ are $C_a$ of 400, 600, and 800 $\mu$mol·mol$^{-1}$. Data represent mean values ± SD.

The simulated $g_m$, calculated by the SWC- and $g_{m,0}$-dependent function (Equation (3)) and the coupled $P_{n,L}$-$g_m$ model (Equation (4)), is presented in Figure 5. Regardless of the model used, the calculated $g_m$ decreased with $C_a$ (Figure 5). Applying Equation (3), the simulated $g_m$ increased almost linearly with an increase in SWC levels and tended to be higher with lower $q_m$, except under excess SWC (25.55% of SWC) (Figure 5a,c,e). In contrast, the simulated $g_m$ calculated by Equation (4), using various $q_m$ values, produced a more complicated tendency to SWC. (Figure 5b,d,f).

The relationships between measured and calculated $g_m$ based on Equations (3) and (4) are shown in Table 2. Both model approaches produced significant relationships between simulated and measured results ($p < 0.05$). Setting the same $q_m$ value, Equation (4) led to a higher $R^2$ (0.44 ~ 0.79) between the estimated and measured results than that of Equation (3) (0.34 ~ 0.52), and the former caused less deviation (0.0055 ± 0.0038 ~ 0.0097 ± 0.0046) from measurements than the latter (0.0090 ± 0.0058 ~ 0.0159 ± 0.0078). Therefore, the proposed coupled $P_{n,L}$-$g_m$ model with well parameterized $q_m$ ($q_m = 0.25$, Equation (4)) effectively improved the predictive accuracy of $g_m$ compared to the previously introduced $g_{m,p}$- and SWC-dependent model (Equation (3)).

### 4.3. Measured and Modeled Instantaneous WUE at Leaf and Whole-Plant Level

At the leaf level, elevated $C_a$ significantly enhanced the measured $WUE_{i,L}$ ($p < 0.05$). Variations in SWC also significantly influenced the measured $WUE_{i,L}$ ($p < 0.05$), which increased as the severe drought was alleviated (SWC increased from 10.48% to 14.41%), followed by a decline with increasing SWC levels and was almost constant when the SWC was above 19.65% (Figure 6a). In both model configurations, the response pattern of simulated $WUE_{i,L}$ to SWC × $C_a$ was similar to that of measured values, except that the simulated $WUE_{i,L}$ increased as the SWC improved from 14.41 to 17.03% at $C_{400}$ and $C_{600}$, departing from the observed decreasing trend (Figure 6a,b).

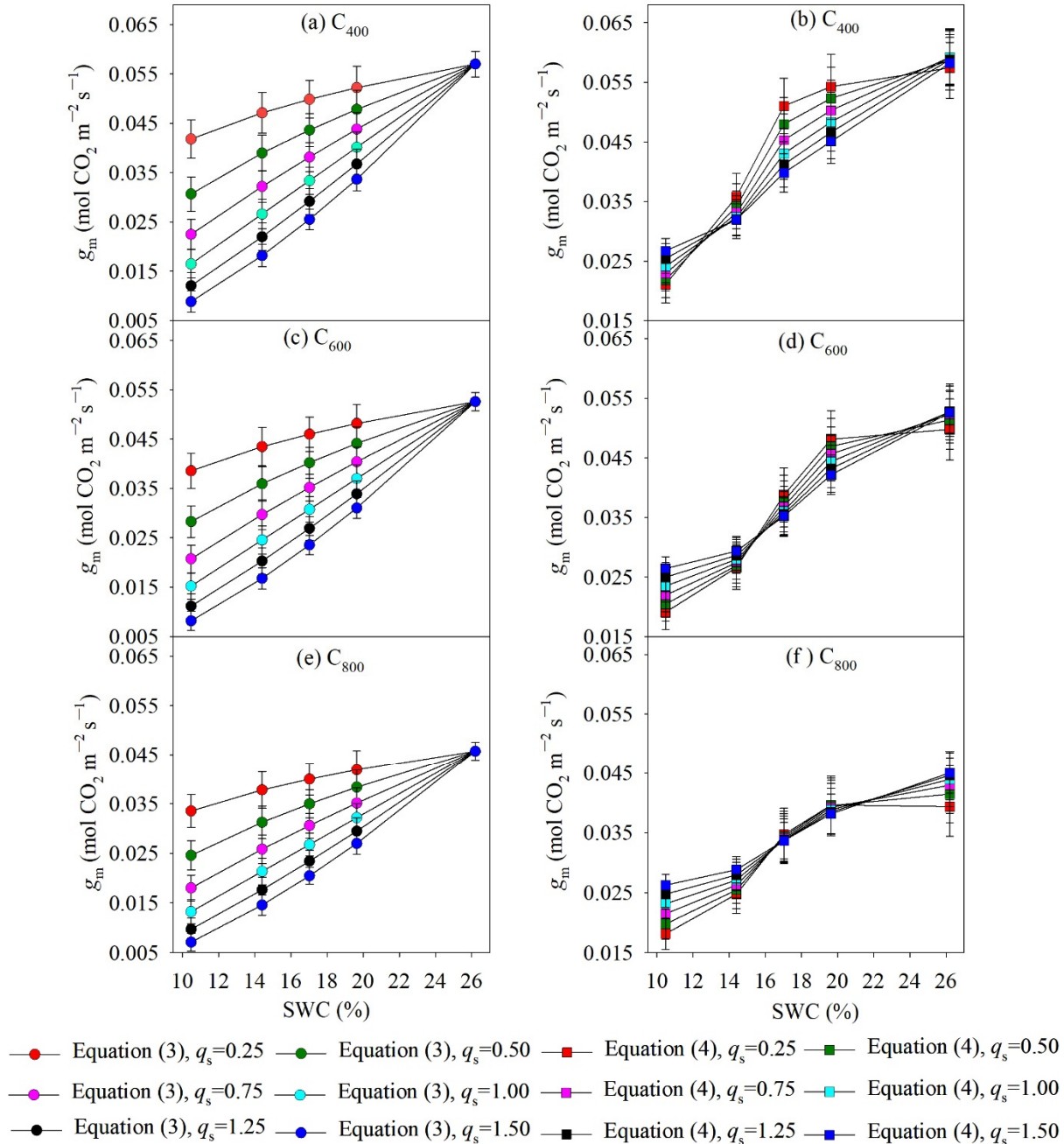

**Figure 5.** The estimated leaf mesophyll conductance ($g_{m}$, mol $CO_2 \cdot m^{-2} \cdot s^{-1}$) is based on different models (Equation (3), (**a**,**c**,**e**); and Equation (4), (**b**,**d**,**f**) under varying soil water contents (SWC) and $CO_2$ concentrations ($C_a$). $C_{400}$, $C_{600}$, and $C_{800}$ are $C_a$ of 400 (**a**,**b**), 600 (**c**,**d**), and 800 µmol·mol$^{-1}$ (**e**,**f**). Tunable parameter $q_m$ is a measure of the nonlinearity of the effects of soil water stress on the mesophyll limiting mechanisms. Data represent mean values ± SD.

At the whole-plant level, it was observed that $C_a$ and SWC significantly influenced ($p < 0.05$) the measured instantaneous WUE (WUE$_{i,P}$). In general, the measured WUE$_{i,P}$ was higher at elevated $C_a$ levels (Figure 6c). When the SWC increased from 10.48% to 14.41%, the percentage increase in the measured WUE$_{i,P}$ was more pronounced at $C_{800}$ than at $C_{400}$ and $C_{600}$. In response to further increases in SWC, the measured WUE$_{i,P}$ generally decreased sharply with further rises in SWC, but this trend was lesser when the soil water status was more than 19.65% of SWC. In both model configurations, the measured and simulated WUE$_{i,P}$ values were similar in their response patterns to SWC × $C_a$, except when the SWC increased from 14.41% to 17.03% at $C_{400}$ and $C_{600}$ (Figure 6c,d).

**Table 2.** Correlation analysis between measured and modeled leaf mesophyll conductance ($g_m$, mol $CO_2$·$m^{-2}$·$s^{-1}$) using different models (Equations (3) and (4)).

| Model | Regression of Measured and Modeled Leaf $g_{sw}$ | | |
| --- | --- | --- | --- |
| | **Linear Regression Equation** | $R^2$ | $p$ |
| Equation (3), $q_m = 0.25$ | $y = 0.30x + 0.03$ | 0.50 | <0.01 |
| Equation (3), $q_m = 0.50$ | $y = 0.44x + 0.02$ | 0.52 | <0.01 |
| Equation (3), $q_m = 0.75$ | $y = 0.53x + 0.02$ | 0.48 | <0.05 |
| Equation (3), $q_m = 1.00$ | $y = 0.58x + 0.01$ | 0.43 | <0.05 |
| Equation (3), $q_m = 1.25$ | $y = 0.61x + 0.01$ | 0.38 | <0.05 |
| Equation (3), $q_m = 1.50$ | $y = 0.61x + 0.03$ | 0.34 | <0.05 |
| Equation (3), $q_m = 0.25$ | $y = 0.79x + 0.01$ | 0.79 | <0.01 |
| Equation (3), $q_m = 0.50$ | $y = 0.72x + 0.01$ | 0.72 | <0.01 |
| Equation (3), $q_m = 0.75$ | $y = 0.65x + 0.01$ | 0.65 | <0.01 |
| Equation (3), $q_m = 1.00$ | $y = 0.57x + 0.02$ | 0.57 | <0.01 |
| Equation (3), $q_m = 1.25$ | $y = 0.51x + 0.02$ | 0.51 | <0.01 |
| Equation (3), $q_m = 1.50$ | $y = 0.44x + 0.02$ | 0.44 | <0.01 |

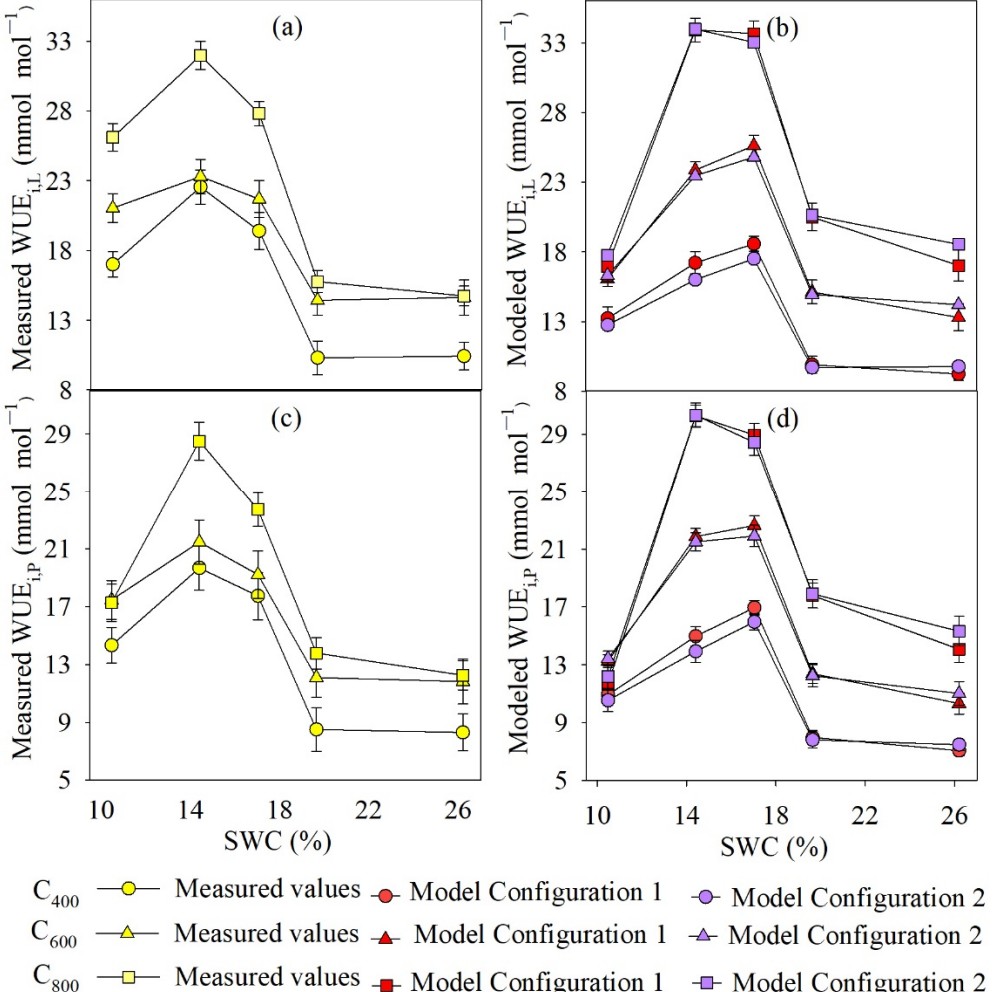

**Figure 6.** Measured (**a**) and simulated (**b**) leaf water use efficiency (WUE$_{i-L}$, mmol·$mol^{-1}$), and measured (**c**) and simulated (**d**) whole-plant level instantaneous water use efficiency (WUE$_{i-P}$, mmol·$mol^{-1}$) in different model configurations under five soil water contents (SWC) × three $CO_2$ concentrations ($C_a$) conditions. $C_{400}$, $C_{600}$, and $C_{800}$ are $C_a$ of 400, 600, and 800 μmol·$mol^{-1}$. Data represent mean values ± SD.

At the leaf and whole-plant level, both models revealed a strong correlation between the measured and calculated instantaneous WUE ($p < 0.01$). However, the relationship was stronger for model configuration 1 (C1), relative to model configuration 2 (C2) (Figure 7). In C1, the calculated $WUE_{i,L}$ ($WUE_{i,P}$) deviated from measured $WUE_{i,L}$ by $3.12 \pm 2.44$ ($2.59 \pm 1.86$) mmol·mol$^{-1}$, which was slightly less than that realized with C2 ($3.14 \pm 2.52$ ($2.62 \pm 1.90$) mmol·mol$^{-1}$ (Figure 7). This indicates that C1 was more accurate than C2 in predicting $WUE_{i,L}$ and $WUE_{i,P}$.

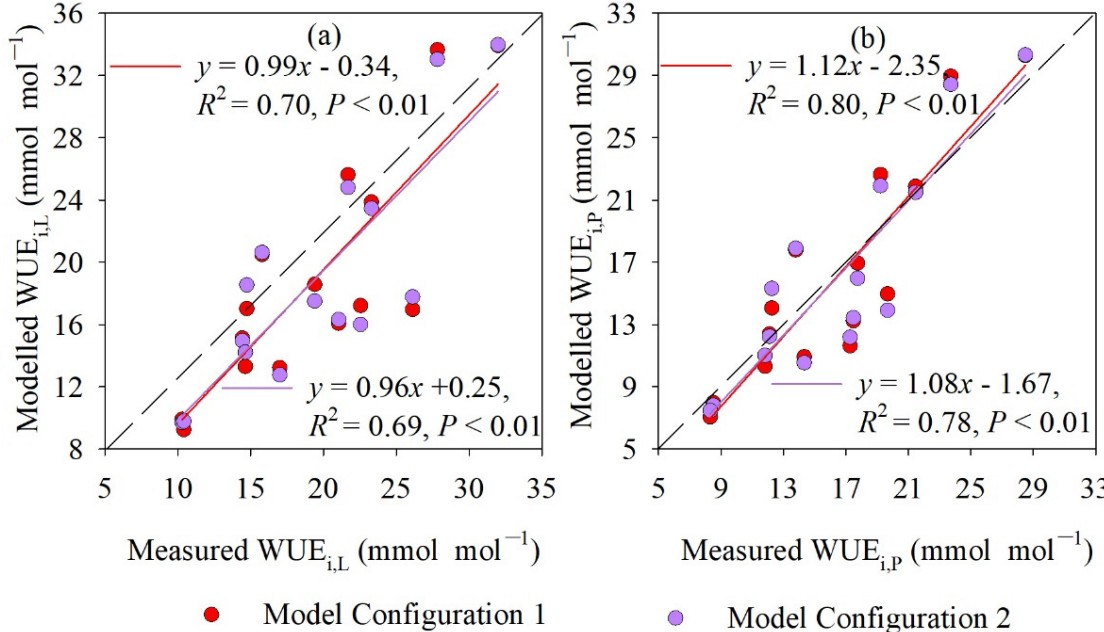

**Figure 7.** Correlation analysis between measured and modeled results of leaf instantaneous water use efficiency ($WUE_{i-L}$, mmol·mol$^{-1}$) estimated by different model configurations (**a**), as well as between measured and modeled results of whole-plant instantaneous water use efficiency ($WUE_{i-P}$, mmol·mol$^{-1}$) estimated by different model configurations (**b**).

*4.4. Comparison of Measured and Modeled $WUE_{s,P}$ Values*

The measured and simulated $WUE_{s,P}$ values are shown in Figure 8. At severe drought (10.48% of SWC), the measured $WUE_{s,P}$ peaked at $C_{600}$ and was lowest at $C_{800}$, whereas the simulated $WUE_{s,P}$, in both model configurations, reached its maximum at $C_{600}$ and was lowest at $C_{400}$ (Figure 8). At an improved soil water status (SWC at 14.41% ~ 25.55%), the measured and simulated $WUE_{s,P}$ values significantly increased due to elevated $C_a$ levels ($p < 0.01$). The measured $WUE_{s,P}$ was also significantly influenced by SWC, generally responding in a similar manner as the measured $WUE_{i,P}$ in response to SWC. When the saplings were subjected to SWC of 14.41% ~ 25.55%, in C1, the response pattern of simulated $WUE_{s,P}$ to SWC was consistent with that of the measured values. In contrast, in C2, the simulated $WUE_{s,P}$ increased as the SWC increased from 19.65 to 26.20% under any $C_a$, which differed from the response pattern of the measured values (Figure 8).

In both model configurations, there was a strong correlation between the measured and calculated $WUE_{s,P}$ ($p < 0.01$). However, the correlation ($R^2$) was stronger for the C1 model, relative to the C2 model (Figure 9). In the C1 model, the calculated $WUE_{s,P}$ deviated from measured $WUE_{s,P}$ by $2.77 \pm 2.23$ mmol·mol$^{-1}$, compared with $2.91 \pm 2.95$ mmol·mol$^{-1}$ for the C2 model (Figure 9). Therefore, compared with C2, C1 better predicts the actual $WUE_{s,P}$.

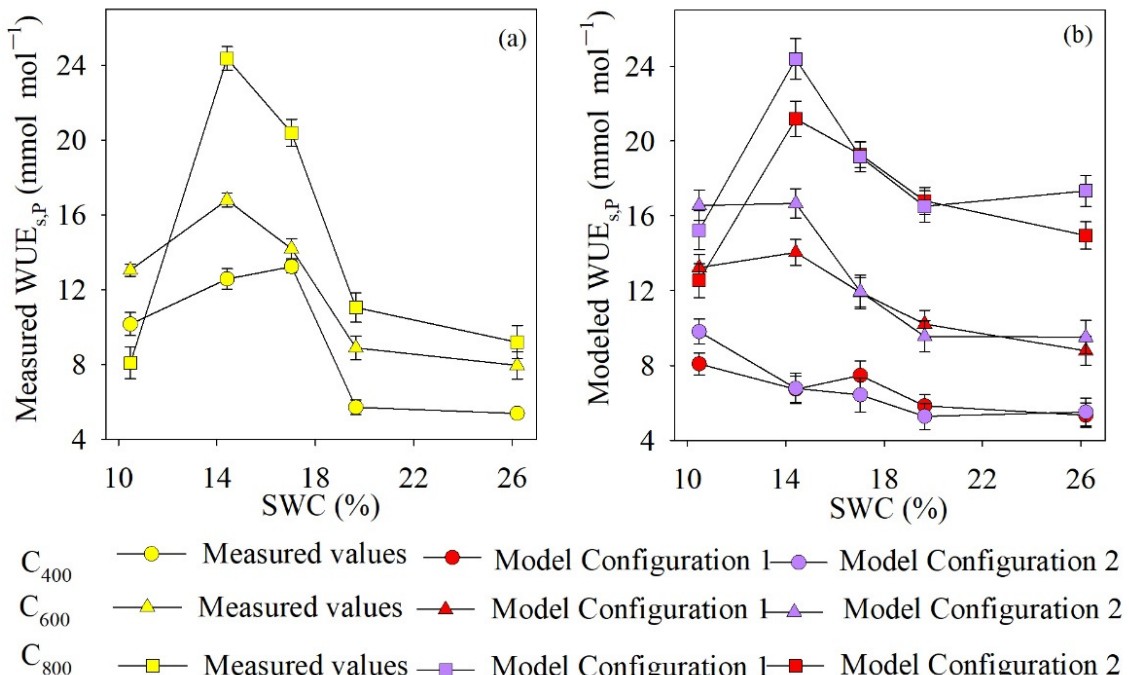

**Figure 8.** Measured (**a**) and modelled (**b**) whole-plant short-term water use efficiency ($WUE_{s,P}$, mmol·mol$^{-1}$) under five soil water contents (SWC) × three $CO_2$ concentrations ($C_a$) conditions. $C_{400}$, $C_{600}$, and $C_{800}$ are $C_a$ of 400, 600, and 800 µmol·mol$^{-1}$. Data represent mean values ± SD.

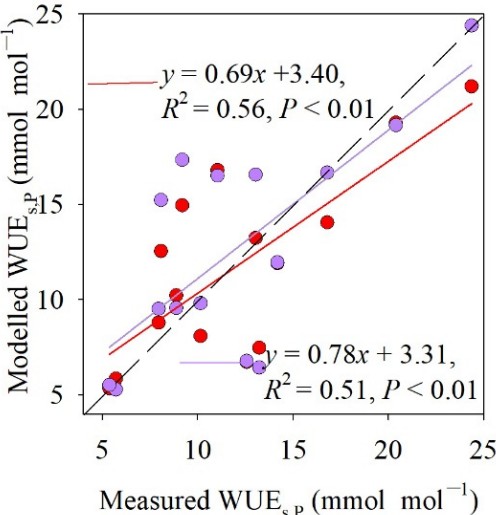

**Figure 9.** Correlation analysis between measured and modeled whole-plant short-term water use efficiency ($WUE_{s,P}$, mmol·mol$^{-1}$) estimated by different model configurations.

## 5. Discussion

### 5.1. Model Performance for Estimating $g_{sw}$ and $g_m$

Soil water stress exclusion in the coupled $P_{n,L}$-$g_{sw}$ model (Equation (1)) for response patterns of $g_{sw}$ performed reasonably well under non-limiting soil water conditions (Table 1), which is in agreement with previous studies conducted in almond trees [22] and in maize and soybean plants [21]. In response to contrasting soil water treatments, the combination of a soil moisture-dependent function with the coupled $P_{n,L}$-$g_{sw}$ model (Equation (2)) using

well parameterized $q_s$ (0.25), was slightly more capable of representing the observed pattern of $g_{sw}$ than Equation (1). These results align with a previous study, which highlights the importance of including the water stress function in the coupled $P_{n,L}$-$g_{sw}$ model [35]. However, adding the soil water stress-dependent function to the coupled $P_{n,L}$-$g_{sw}$ model contributed little to the improvement of model performance for $g_{sw}$. This can be ascribed to the fact that the measured $P_{n,L}$ incorporated the effect of the soil water status, resulting in the estimation of $g_{sw}$ from the coupled $A_n$-$g_{sw}$ model accounting for the effect of soil water.

The Keenan et al. (2010) [12] model (Equation (3)) for $g_m$ was insufficient to take into account the impact of $C_a$ and was therefore less suitable to simulate $g_m$ (Table 2). In contrast, the predictive accuracy improved considerably when estimating $g_m$ using the coupled $P_{n,L}$-$g_m$ model (Equation (4)). Therefore, the proposed coupled $P_{n,L}$-$g_m$ model is valid and promising for simulating $g_m$, despite its phenomenological nature and dependence on physiological hypotheses. Furthermore, imposing $q_m = 0.25$ in Equation (4) provided the best fit with the measured values (Figure 6), indicating that the limitation strength of $g_m$ was similar to that of $g_{sw}$. This result conflicts with the general findings that stomatal behaviour imposed a higher limitation on photosynthesis than mesophyll behavior [22,36,37]. However, such a phenomenon may not always occur. For example, Pérez-Martín et al. (2009) [4] observed minor difference between stomatal and mesophyll limitations and reported that stiffer and more sclerophyllous leaves would provide greater mesophyll resistance during $CO_2$ diffusion.

In addition, this study found that, mostly, $g_{sw}$ (and $g_m$) values varied with SWC, even if the influence of $C_a$ on $g_{sw}$ (and $g_m$) was significant. Centritto et al. (2002) [38] also found that $g_{sw}$ was significantly lower in water-stressed seedlings than in well-watered seedlings, while elevated $C_a$ did not significantly influence $g_{sw}$ under either well-watered or water-stressed conditions. However, Flexas et al. (2007) [24] observed that both $g_{sw}$ and $g_m$ were much higher in 400 $\mu mol \cdot mol^{-1}$ air than those in 1000 $\mu mol \cdot mol^{-1}$ air. Thus, $C_a$ effects on $g_{sw}$ (or even $g_m$) may not be universal across species.

### 5.2. Different Model Configurations for Estimating $WUE_{s,P}$

In our proposed short-term WUE model (Equation (15)), scaling up from the leaf to the whole-plant level, there are diffusive limitation parameters. The C1 inferred from Equations (2) and (4) could more accurately represent the observed $WUE_{s,P}$ than the C2 inferred from Equations (1) and (3) (Figure 9). This leads us to infer that the model scaling up from the leaf to whole-plant level, based on more accurate stomatal and mesophyll behaviour predictions, could be used to estimate $WUE_{s,P}$ with a high level of precision. In addition, the developed model for estimating $WUE_{s,P}$ also contains photosynthetic parameters by introducing the coupled $P_{n,L}$-$g_{sw}$ and $P_{n,L}$-$g_m$ models. Rather than estimating $P_{n,L}$ via the photosynthesis model [21,39], the estimates of $WUE_{s,P}$ were calculated from measured $P_{n,L}$ values to exclude the situation that errors in the representation of $g_{sw}$ and $g_m$ might be compensated or overwhelmed by errors in simulated $P_{n,L}$. In such a situation, we can identify the influence of precision of stomatal and mesophyll modelling on the credibility and accuracy of the developed $WUE_{s,P}$ model.

For $WUE_{i,L}$ and $WUE_{i,P}$ modelling, the C1 inferred from the more accurate $g_{sw}$ model incorporating a soil water stress-dependent function (Equation (2)) slightly outperformed the C2 (Figure 7a,b). However, the $R^2$ between measured and modelled $WUE_{s,P}$ were lower than those of $WUE_{i,L}$ and $WUE_{i-P}$ (Figures 7 and 9), most likely because the involvement of more parameters in the isotope-inferred $WUE_{s,P}$ model could introduce more uncertainties and errors. For example, complications arising from post-photosynthetic carbon isotope fractionations are not considered as the process is still difficult to assess and largely unknown [40,41]. Furthermore, the effects of photorespiration and mitochondrial respiration on photosynthetic $^{13}C$ discrimination are still the subject of debate [25] and were thus ignored in the current study.

*5.3. Uncertainties of WUE$_{s,P}$ Introduced by g$_{sw}$ and g$_m$*

Uncertainty analysis was conducted to further determine the uncertainties of WUE$_{s,P}$ associated with stomatal and mesophyll behaviour simulations. Using the most effective approach to reproduce $g_{sw}$ (Equation (2), with tunable parameter $q_s$ = 0.25) and $g_m$ (Equation (4), with tunable parameter $q_m$ = 0.25), the average uncertainties (s.d.) in $g_{sw}$ and $g_m$ were 17.10% (14.14%) and 15.39% (10.98%), respectively. The WUE$_{s,P}$ estimated from C1 caused average uncertainties (s.d.) of 24.09% (21.61%). The relatively small discrepancies between mean value and standard deviation in uncertainties of $g_{sw}$, $g_m$, and WUE$_{s,P}$ indicate that these estimation methods were not stable, although model performance was improved. In addition, the WUE$_{s,P}$ was more sensitive to $g_{sw}$ than to $g_m$. That is, 10% error in $g_{sw}$ introduced 6.17% error in WUE$_{s,P}$, while 10% error in $g_m$ introduced a smaller error of 4.48% in WUE$_{s,P}$. Although the stomatal and mesophyll limitations were similar to those of the photosynthetic process in this study, the leaf transpiration is exclusively controlled by $g_{sw}$ when the $v$ is almost constant [9,10,42] (Seibt et al., 2008; Zhao et al., 2017; Zhang et al., 2019), which could result in the $g_{sw}$ being a more influential factor for WUE$_{s,P}$ than the $g_m$.

Overall, the explored whole-plant model, based on well-characterized coupled $P_{n,L}$-$g_{sw}$ (Equation (2)) and $P_{n,L}$-$g_m$ models (Equation (4)), is applicable for evaluating variation in WUE$_{s,P}$ in response to $C_a$ and SWC. However, we recognize that *Platycladus orientalis* is a very specific plant, and the results are hard to generalize for all other plants. It is therefore important to collect data from different plants to further examine the model. In addition, using only data of pot-grown saplings acclimated in growth chambers, with relatively similar canopy components (canopy structure, light interception), is not convincing enough for a general verification of the developed modelling approach. For field-grown plants with a complex canopy structure, water potential and gas exchange information for individual leaves cannot be consistent for the whole-plant level [43,44], leading to difficulties in generalizing the estimation of WUE$_{s,P}$ from leaf properties. Moreover, root systems have been excluded from gas exchange measurements due to it being impossible to separate root and soil respiration for technical restriction. In conclusion, the ability of the whole-plant model to simulate WUE$_{s,P}$ features should still be explored and improved.

## 6. Conclusions

In this study, the performances of coupled $P_{n,L}$-$g_{sw}$ and $P_{n,L}$-$g_m$ models were evaluated using leaf gas exchange measurements. We found the coupled $P_{n,L}$−$g_{sw}$ model incorporating the water stress-dependent function with well parameterized $q_s$ (Equation (2)) agreed slightly better with the measured $g_{sw}$ values than the model excluding the soil water stress effect (Equation (1)), and the established coupled $P_{n,L}$-$g_m$ model with well parameterized $q_m$ (Equation (4)) allowed for a more reliable estimation of $g_m$ than the previously introduced $g_{m,P}$- and SWC-dependent model (Equation (3)). Based on the well-characterized models describing stomatal and mesophyll behavior, an isotopic model, scaling from the leaf to whole-plant level for estimating WUE$_{s,P}$ (Equation (16)), was then established and validated. We found the developed model for WUE$_{s,P}$ proved effective at capturing response patterns to $C_a$ and SWC. Therefore, introducing the model performing well for $g_{sw}$ and $g_m$ into the Farquhar et al. (1989) model was applicable and represents a promising approach for describing whole-plant WUE at smaller temporal scales.

**Author Contributions:** Y.Z. (Yonge Zhang) and B.L. designed and performed the experiment. Y.Z. (Yonge Zhang) analyzed the data and wrote the manuscript. G.J. revised the paper and finished the submission. X.Y. (Xinxiao Yu) and X.Y. (Xiaolin Yin) contributed significantly to data analysis. X.Z. contributed to funding acquisition. Y.Z. (Yang Zhao), Z.W. and C.C. contributed to the manuscript preparation and language edit. Y.W. and Y.X. contributed to the practice of the experiment. All authors have read and agreed to the published version of the manuscript.

**Funding:** This research was funded by the National Natural Science Foundation of China (No. 51979290, 51879281 and 32001372), and the Ningxia Water Conservancy Science and Technology Project (SBZZ-J-2021-13 and SBZZ-J-2021-12).

**Conflicts of Interest:** The authors declare no conflict of interest.

## Abbreviations

The following abbreviations are used in this manuscript:

| | |
|---|---|
| SSWC | Soil water content |
| $\theta$ (SWC) | Actual soil water content |
| $\theta_c$ | Soil water content at field capacity (26.20%) |
| $\theta_w$ | Soil water content at permanent wilting point (4.08%) |
| $C_a$ | Atmosphere $CO_2$ concentration ($\mu mol \cdot mol^{-1}$). The $C_{600}$ and $C_{800}$ are $C_a$ levels of 600 $\mu mol \cdot mol^{-1}$ and 800 $\mu mol\ mol^{-1}$, and the $C_{400}$ is $C_a$ level of 400 $\mu mol \cdot mol^{-1}$. |
| WUE | Water use efficiency ($mmol \cdot mol^{-1}$) |
| $WUE_{i\text{-}L}$ | Leaf instantaneous water use efficiency ($mmol \cdot mol^{-1}$) |
| $WUE_{i\text{-}P}$ | Whole-plant instantaneous water use efficiency ($mmol \cdot mol^{-1}$) |
| $WUE_{s\text{-}P}$ | Whole-plant short-term water use efficiency ($mmol \cdot mol^{-1}$) |
| $P_{n,L}$ | Leaf daytime net photosynthetic rate ($\mu mol \cdot m^{-2} \cdot s^{-1}$) |
| $E_L$ | Leaf daytime transpiration rate ($mmol \cdot m^{-2} \cdot s^{-1}$) |
| $P_{n,P}$ | Whole-plant daytime net photosynthetic rate ($mmol \cdot h^{-1}$) |
| $\int P_{n,P}$ | Whole-plant cumulative net carbon sequestration over a day-night cycle ($mmol^{-1}$) |
| $E_P$ | Whole-plant daytime transpiration rate ($mol \cdot h^{-1}$) |
| $E_d$ | Whole-plant nighttime transpiration rate ($mol \cdot h^{-1}$) |
| $\int E_P$ | Whole-plant cumulative transpiration over a day-night cycle ($mol^{-1}$) |
| $R_P$ | Whole-plant nighttime respiration rate ($mmol \cdot h^{-1}$) |
| $C_s$ | Leaf surface $CO_2$ concentration ($\mu mol \cdot mol^{-1}$) |
| $C_i$ | Leaf intercellular $CO_2$ concentration ($\mu mol \cdot mol^{-1}$) |
| $g_b$ | Leaf boundary layer conductance ($mol\ CO_2 \cdot m^{-2} \cdot s^{-1}$) |
| $g_{sw}$ | Leaf stomatal conductance ($mol\ H_2O \cdot m^{-2} \cdot s^{-1}$) |
| $g_{sc}$ | Leaf stomatal conductance for $CO_2$ ($mol\ CO_2 \cdot m^{-2} \cdot s^{-1}$) |
| $g_1$ | Fitted parameter associated with the photosynthesis–stomatal conductance model |
| $g_{0,sw}$ | Fitted parameter, and $g_{0,sw}$ is considered to represent the residual stomatal conductance ($mol\ H_2O \cdot m^{-2} \cdot s^{-1}$) |
| $f(\theta_s)$ | Stomatal conductance limitation function that depends on soil water stress |
| $q_s$ | The exponents involved in the stomatal conductance limitation function |
| $g_m$ | Leaf mesophyll conductance ($mmol\ CO_2 \cdot m^{-2} \cdot s^{-1}$) |
| $g_{m,p}$ | Potential (unstressed) $g_m$ ($mmol\ CO_2 \cdot m^{-2} \cdot s^{-1}$) |
| $g_2$ | Fitted parameter associated with the photosynthesis–mesophyll conductance model |
| $g_{m,0}$ | Fitted parameter, and $g_{0,m}$ is considered to represent the residual mesophyll conductance ($mol\ CO_2 \cdot m^{-2} \cdot s^{-1}$) |
| $f(\theta_m)$ | Mesophyll conductance limitation function that depends on soil water stress |
| $q_m$ | The exponents involved in the mesophyll conductance limitation function |
| $\Delta_{mea}$ | Measured short-term photosynthetic $^{13}C$ discrimination (‰) |
| $\Delta_{lin}$ | The $^{13}C$ discrimination calculated by the linear model (‰) |
| $\delta^{13}C_a$ | The $\delta^{13}C$ of atmosphere $CO_2$ (‰) |
| $\delta^{13}C_l$ | The $\delta^{13}C$ of leaf water-soluble organic materials (WSOM) (‰) |
| $a$ | Fractionation associated with the $CO_2$ diffusion in air (4.4‰) |
| $b'$ | Fractionation relevant to the reactions of Rubisco and PEP carboxylase (27‰) |
| $a_m$ | Fractionation of $CO_2$ diffusion and dissolution in the liquid phase (1.8‰) |
| $a_i$ | Fractionation of $CO_2$ diffusion and dissolution in the liquid phase (1.8‰) |
| $b$ | Fractionation during carboxylation (29‰) |
| $e$ | Discrimination value for the mitochondrial respiration (dark respiration) |
| $f$ | Discrimination value for photorespiration |

| | |
|---|---|
| $\Gamma$ | $CO_2$ compensation point with dark respiration |
| $k$ | Carboxylation efficiency |
| $D$ | Water vapor pressure difference between the intercellular spaces of the leaf and the leaf external air (mbar) |
| $\phi_{w,i}$ | Instantaneous proportion of "unproductive" water loss, that is, water lost by transpiration from twigs and stems during the day |
| $\phi_{c,i}$ | Instantaneous proportion of carbon fixed during photosynthesis, that is, subsequently lost by respiration from twigs and stems during the day |
| $\phi_{w,s}$ | Proportion of "unproductive" water loss at short time scale (over a day–night cycle), that is, water lost by transpiration from twigs and stems during the day, and from twigs, stems, and leaves at night |
| $\phi_{c,s}$ | Proportion of carbon fixed during photosynthesis at short time scale (over a day–night cycle), that is, subsequently lost by respiration from twigs and stems over the whole period, and from leaves during the night |
| $LA$ | Total leaf area ($m^2$) |
| $DW$ | Dry weight (g) |

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
