# Peer review of "Scaling Up from Leaf to Whole-Plant Level for Water Use Efficiency Estimates Based on Stomatal and Mesophyll Behaviour in Platycladus orientalis"

_water, doi:10.3390/w14020263_

Round 1
Reviewer 1 Report
I read the manuscript of Zhang et al., the authors made an excellent study on WUE models, adding some new information on the role of mesophyll conductance both on photosynthesis and transpiration. The work has been clearly explained and responds to the objectives initially proposed.
The methodology is adequate. The results are clearly stated and have been discussed taking into account relevant works in this area. The bibliography has been adequately selected, including the most current and relevant.
I do not consider that modifications have to be made in the structure
of the text.
Some minor revision are report in the attached file.

Author Response
Dear Reviewer:
Thank you very much for your attention and the reviewers’ evaluation and comments on our paper “Scaling-up from leaf to whole-plant level for water use efficiency estimates based on stomatal and mesophyll behavior in Platycladus orientalis”. We have learned much from the comments which are valuable and helpful for revising and improving our manuscript, as well as the important guiding significance to our research. After carefully studying the comments and making corresponding correction, we have made corresponding changes. Revise portion are marked in red in the manuscript. The main corrections in the paper and responses to the reviewer’ comments are as following:
- The term loss is inappropiate. Efficiency is normally a ratio between output/input. As an example I would suggest the use of absorbed by the roots.
Response: The term water loss is widely used by other researchers, for example, Cernusak LA, Aranda J, Marshall JD, Winter K (2007) Large variation in whole-plant water-use efficiency among tropical tree species. New Phytol 1732:294–305.
The term water loss can be defined at different spatial scales. At the leaf level, water loss is the leaf transpiration, including daytime and nighttime transpiration. At the whole-plant level, whole parts (leaf, twigs and stem) transpires water during daytime and nighttime (Page 5, line 152-155).
- This process take into account primary and secondary metabolism of the plants.
Response: The whole-plant respiration describes respiration of photosynthetic and non-photosynthetic parts. In this study, root systems have been excluded from gas exchange measurements, due to it was impossible to separate root and soil respiration for technical restriction (Page 5, line 152-155).
- Please define better what is water loss
Response: Considering the reviewer’s suggestion, we have define better what water loss is (Page 1, line 25).
- night transpiration?
Response: At the whole-plant level, “unproductive” water loss includes transpiration from non-photosynthetic parts (twigs and stem) during the daytime, and that from whole parts (leaf, twigs and stem) during the nighttime.
- Ca is ?
Response: The Ca is CO2 concentration. Considering the reviewer’s suggestion, we have completed names when using abbreviations for the first time. In addition, we have added a table of important notation to a supplementary material in page 23, line 561.
- evaporation? or water used in the respiration metabolism?
Response: That is twigs and stem transpiration.

Reviewer 2 Report
The submitted Manuscript provides interesting results on the important topic (water use efficiency by plants, skilled combination of experimental results and modelling, potential impact of global climate change on plants).
However, several points are to be addressed and the MS has to be amended.
1) Reference 19, misspelling. Pls, check all the text.
- Jarvis, P. G. The interpretation of the variations in leaf water potential and stomatal conductance found incCanopies in the 594 field.
2) Pls, check all the references and correct them according to the journal style.
3) Generally speaking, the short-term water use efficiency for individual leaves can not be generalized for whole plant level. Please, at least consider reading the paper:
McCutchan, H., Shackel, K. A. (1992). Stem-water Potential as a Sensitive Indicator of Water Stress in Prune Trees (“Prunus domestica“ L. cv. French). J. Am. Soc. Hortic. Sci. 117, 607–611. doi: 10.21273/JASHS.117.4.607
Moreover, there are publications that water potential varies along leaves depending on many factors. It is reasonable to mention and discuss the points.
4) The Reviewer is basically impressed by the part:
The soil water stress-dependent limitation functions (f(θi)) were provided previously 129 (Keenan et al., 2010; Porporato et al., 2001): 130
, (5) 131 i c w i w c c w w 1 ( ) 0 q f
where θ is the actual soil water content (%); θc and θw are soil water content levels at 132 field capacity (26.20%) and permanent wilting point (4.08%), respectively; and the sub- 133 script i = s and m represent stomatal and mesophyll limitations, respectively. In this study, 134Water 2021, 13, x FOR PEER REVIEW 4 of 20
the selected values for tunable parameters of qs and qm were 0.25, 0.50, 0.75, 1.00, 1.25, and 135 1.50, within the previously reported range (Egea et al., 2011b; Keenan et al., 2010). 136
The wilting point and the field capacity evidently depend on the type of soil and the plants studied where both input parameters provide nonlinearity. A simple online search shows, for example, the typical curves: figure 1 from https://extension.okstate.edu/fact-sheets/understanding-soil-water-content-and-thresholds-for-irrigation-management.html
Please, describe the type of the soil used with more characteristics. Typically, soil water potential is measured by tensiometers; the parameter of soil matrix potential is important for growth and water uptake of plants. Are there any curves for soil used in the experiments?
5) Platycladus orientalis is a very specific plant when the results are hardly generalized for all the other plants. It is the evergreen coniferous species with presumably specific stomatal regulation while most plants (at least used in agriculture) are monocots and dicots.
Pls, add then the name of the plant to the title to avoid unnecessary conclusions.
6) The controlled environment (light, air temperature, and relative humidity) in the 213 growth chambers was set to simulate natural growth conditions.
Please, indicate the conditions of illumination, the type of lamps/LEDs used or the spectral composition of light used.
7) and subjected to a nested de-211 sign with three Ca levels and five SWC regimes.
Please, use complete names when using abbreviations for the first time.
8) For the sake of calculative simplicity, we 222 assumed that the SWC gradient was: 1) 10.48%, 2) 14.41%, 3) 17.03%, 4) 19.65%, and 5) 223 26.20%, respectively. To reduce soil evaporation, the surface of the potting soil was cov-224 ered with an approximately 2-cm layer of perlite. The desired SWC regimes were achieved 225 following the method from Zhang et al. (2019).
- a) How often the plants were watered and what was the method for keeping so stable SWC?
- b) Again, it’s important to describe the soil due to nonlinearity of curves for soil water potential against SWC.
9) On the day of whole-plant carbon balance measurements, leaf gas change properties 267 (Pn,L, EL, gs, and Ci), leaf temperature (TL), and leaf surface relative humidity (RH) were 268 measured inside the growth chambers on fully expanded leaves, using an open gas ex-269 change system (Li-6400, Li-Cor, Lincoln, NE, USA).
What was considered as fully expanded leaves for the coniferous species?
10) 3.2.4. Whole-plant total leaf area measurement
How was the leaf area measured for the complex leaves (for the subsample)? Was it the same ratio of leaf area for different treatments?
11) To realize orthogonal treatments, two growth chambers were used: one with 217 ambient Ca (400 μmol·mol-1) and another one with elevated Ca (600 and 800 μmol·mol-1) 218 levels.
How was it experimentally possible to ensure 2 levels of CO2 (600 and 800 μmol·mol-1) in one chamber?
12) Measured gm was obtained by 288 carbon isotope discrimination combined with gas exchange measurements as described 289 by Zhang et al. (2019). 290
Please, describe more how the parameter was measured.
13) Please, indicate the units for gm for figure legends and at the axes of the figures, was it measured as mol CO2·m-2·s-1?
In Variation in mesophyll conductance among Australian wheat genotypes Functional Plant Biology, 2014, 41, 568–580
http://dx.doi.org/10.1071/FP13254 another units are provided.
14) where gm,0 and g2 are fitted parameters and g0,m is considered to represent the residual 127 mesophyll conductance (mol CO2·m-2·s-1).
Pls, correct to g0,m for both.
15) Figure 2, it’s logical to start from model 1 in the figure legend.
16) In contrast, when the effect of soil water 323 stress was incorporated in the coupled Pn,L-gsw model (Eq. 2), the simulated gsw generally 324 increased as SWC increased, regardless of the value imposed by qs. More specifically, at 325 any given Ca, the amplification in simulated gs
Please, indicate what the qs is and how it is expressed. It is not given in equation (2).
17) Figure 3. Same as 15).
18) Figure 4. Pls, indicate where the results di differ statistically. Same for all the figures.
19) Figure 5. What are the parts a-f of the figure? Pls, indicate. The same for different values of qs.
20) Figures 6-10 are heavily overloaded with curves and data, some points are overlapping and not visible.
21) Figure 7, it seems better to divide and present separately the measured and modelled results.
22) Basically it is not clear which models are considered and where.
23) The reasonable ideas and data are not presented well for a reader to understand clearly and potentially reproduce the results. It potentially (the Reviewer is not suggesting a single solution) might be reasonable to move some models to methodical part or to a supplementary material while transforming some figures to tables. So far, the submitted MS is not well presented and requires more information about the soil and illumination used for experiments.
Author Response
Dear Reviewer:
Thank you very much for your attention and the reviewers’ evaluation and comments on our paper “Scaling-up from leaf to whole-plant level for water use efficiency estimates based on stomatal and mesophyll behavior in Platycladus orientalis”. We have learned much from the comments which are valuable and helpful for revising and improving our manuscript, as well as the important guiding significance to our research. After carefully studying the comments and making corresponding correction, we have made corresponding changes. Revise portion are marked in red in the manuscript. The main corrections in the paper and responses to the reviewers’ comments are as following:
- Reference 19, misspelling. Pls, check all the text.
- Jarvis, P. G. The interpretation of the variations in leaf water potential and stomatal conductance found incCanopies in the field.
Response: We are very sorry for our incorrect writing. We have corrected the misspelling (Page 26, line 621) and check all the text.
- Pls, check all the references and correct them according to the journal style.
Response: Considering the reviewer’s suggestion, we have checked all the references and changed them to consistent formatting throughout.
- Generally speaking, the short-term water use efficiency for individual leaves can not be generalized for whole plant level. Please, at least consider reading the paper:
McCutchan, H., Shackel, K. A. (1992). Stem-water Potential as a Sensitive Indicator of Water Stress in Prune Trees (“Prunus domestica“ L. cv. French). J. Am. Soc. Hortic. Sci. 117, 607–611. doi: 10.21273/JASHS.117.4.607
Moreover, there are publications that water potential varies along leaves depending on many factors. It is reasonable to mention and discuss the points.
Response: Considering the reviewer’s suggestion, we have discussed the points in Page 23, line 537-540. We acknowledge the shortcomings of our research. For field-grown plants with a complex canopy structure, water potential and gas exchange information for individual leaves can’t be consistent for whole plant level, leading to difficulties in generalizing the estimation of WUEs,P from leaf properties.
- The Reviewer is basically impressed by the part:
The soil water stress-dependent limitation functions (f(θi)) were provided previously (Keenan et al., 2010; Porporato et al., 2001):
,
where θ is the soil volumetric water content (%); θc and θw are soil water content levels at field capacity (26.20% ) and permanent wilting point (4.08%), respectively; and the subscript i = s and m represent stomatal and mesophyll limitations, respectively. In this study, the selected values for tunable parameters of qs and qm were 0.25, 0.50, 0.75, 1.00, 1.25, and 1.50, within the previously reported range (Egea et al., 2011b; Keenan et al., 2010).
The wilting point and the field capacity evidently depend on the type of soil and the plants studied where both input parameters provide nonlinearity. A simple online search shows, for example, the typical curves: figure 1 from https://extension.okstate.edu/fact-sheets/understanding-soil-water-content-and-thresholds-for-irrigation-management.html
Response: Considering the reviewer’s suggestion, we have described the soil type, and the measurements of wilting point and the field capacity of soil and the plants in Page 6-7, line 190-196. The soil type is sandy loam, and the field capacity (θc, 26.2%) and permanent wilting point (θw, 4.08%) of the soil and plants were determined by a pilot experiment. The θc was measured by soil water content (SWC) sensors (Onset. USA, HOBO–U30) after soil samples absorbing water for 24 h and no vertical underwater droplets. The θw was measured by the sensors (Onset. USA, HOBO–U30) when leaves produced wilting and can’t be restored by supplemental water, that is, below the wilting point leaf water potential (measured by portable plant water potential meter (WP4C, Decagon, Washington, USA); data not shown) didn’t increase with the increase of SWC.
- Please, describe the type of the soil used with more characteristics. Typically, soil water potential is measured by tensiometers; the parameter of soil matrix potential is important for growth and water uptake of plants. Are there any curves for soil used in the experiments?
Response: Considering the reviewer’s suggestion, we have described the soil type in Page 6, line 190. The soil type is sandy loam. Soil water characteristics curves were not discussed in this study, instead we measured the field capacity and permanent wilting point by a pilot experiment using soil water content sensors (Onset. USA, HOBO–U30).
- Platycladus orientalis is a very specific plant when the results are hardly generalized for all the other plants. It is the evergreen coniferous species with presumably specific stomatal regulation while most plants (at least used in agriculture) are monocots and dicots.
Pls, add then the name of the plant to the title to avoid unnecessary conclusions.
Response: Considering the reviewer’s suggestion, we have add the name of the plant to the title to avoid unnecessary conclusions in Page 1, line 2. In addition, we have discussed the deficiency of the manuscript in Page 23, line 532-534. We recognize that Platycladus orientalis is a very specific plant when the results are hardly generalized for all the other plants. It is therefore important to collect data of other plants to further examine the model.
- The controlled environment (light, air temperature, and relative humidity) in the growth chambers was set to simulate natural growth conditions.
Please, indicate the conditions of illumination, the type of lamps/LEDs used or the spectral composition of light used.
Response: Considering the reviewer’s suggestion, we have indicate the conditions of illumination and the type of lamps/LEDs used in Page 7, line 202-209. From 07:00 to 19:00 (simulating daytime), all LED lights were turned on, with 60% relative humidity and 25°C. From 19:00 to 07:00 (simulating nighttime), all LED lights were turned off, with 80% relative humidity and 18°C. In North China, P. orientalis saplings are generally grown under the forest canopy, which receives a lower photosynthetic photon flux density (with an average of 230±37 μmol·m−2·s−1) than full sunlight (with an average of 350±41 μmol·m−2·s−1) at daytime during the growing season. Thus, the low level of light intensity in the growth chamber (220±20 μmol·m−2·s−1) was considered to be approximately appropriate to simulate the growth of understory saplings.
- and subjected to a nested de-211 sign with three Ca levels and five SWC regimes.
Please, use complete names when using abbreviations for the first time.
Response: Considering the reviewer’s suggestion, we have completed names when using abbreviations for the first time in Page 7, line 192 and 212.
- For the sake of calculative simplicity, we assumed that the SWC gradient was: 1) 10.48%, 2) 14.41%, 3) 17.03%, 4) 19.65%, and 5) 223 26.20%, respectively. To reduce soil evaporation, the surface of the potting soil was covered with an approximately 2-cm layer of perlite. The desired SWC regimes were achieved 225 following the method from Zhang et al. (2019).
- a) How often the plants were watered and what was the method for keeping so stable SWC?
- b) Again, it’s important to describe the soil due to nonlinearity of curves for soil water potential against SWC.
Response: Considering the reviewer’s suggestion, we have described that how often the plants were watered and what was the method for keeping so stable SWC in Page 7, line 220-222. The SWC in the upper 10 to 15 cm was continuously measured by sensors (Onset. USA, HOBO–U30), and the water status of each potting soil was checked twice a day and irrigated manually to achieved target SWC regimes.
In addition, we have described the soil type in Page 6, line 190. The soil type is sandy loam.
- On the day of whole-plant carbon balance measurements, leaf gas change properties (Pn,L, EL, gs, and Ci), leaf temperature (TL), and leaf surface relative humidity (RH) were measured inside the growth chambers on fully expanded leaves, using an open gas ex-change system (Li-6400, Li-Cor, Lincoln, NE, USA).
What was considered as fully expanded leaves for the coniferous species?
Response: We are very sorry for our incorrect writing. We have changed “…were measured inside the growth chambers on fully expanded leaves” to “…were measured inside the growth chambers on mature leaves”.
- 2.4. Whole-plant total leaf area measurement
How was the leaf area measured for the complex leaves (for the subsample)? Was it the same ratio of leaf area for different treatments?
Response: Considering the reviewer’s suggestion, we have described that how the leaf area measured for the complex leaves in Page 10, line 294-295. The leaf area for subsample (LAsub) was determined using image–processing software for Photoshop. The whole-plant total leaf area of each sapling was calculated by Eq.( 22) in the manuscript.
- To realize orthogonal treatments, two growth chambers were used: one with ambient Ca (400 μmol·mol-1) and another one with elevated Ca (600 and 800 μmol·mol-1) levels.
How was it experimentally possible to ensure 2 levels of CO2 (600 and 800 μmol·mol-1) in one chamber?
Response: Considering the reviewer’s suggestion, we have described that how it experimentally possible to ensure 2 levels of CO2 (600 and 800 μmol·mol-1) in one chamber in Page 7, line 210-215. To realize orthogonal treatments, two growth chambers were used. One growth chamber was connected to a CO2 tank, which was used to maintain elevated CO2 concentrations (Ca) of 600 μmol·mol−1 and 800 μmol·mol−1. Another growth chamber (b) was used to maintain Ca of approximately 400 μmol·mol−1. The control system in growth chamber can continuously adjust Ca steady near the enactment value with standard deviation of 50 μmol•mol-1.
- Measured gm was obtained by carbon isotope discrimination combined with gas exchange measurements as described by Zhang et al. (2019).
Please, describe more how the parameter was measured.
Response: Considering the reviewer’s suggestion, we have described that how measured gm was obtained by carbon isotope discrimination combined with gas exchange measurements in Page 10, line 284-290.
- Please, indicate the units for gm for figure legends and at the axes of the figures, was it measured as mol CO2m-2·s-1?
In Variation in mesophyll conductance among Australian wheat genotypes Functional Plant Biology, 2014, 41, 568–580 http://dx.doi.org/10.1071/FP13254 another units are provided.
Response: Considering the reviewer’s suggestion, we have indicated the units for gm (mol CO2·m-2·s-1) for figure legends and at the axes of the figures. In addition, we have checked all the text and corrected them.
- where gm,0 and g2 are fitted parameters and g0,m is considered to represent the residual mesophyll conductance (mol CO2m-2·s-1).
Pls, correct to g0,m for both.
Response: We are very sorry for our incorrect writing. We have changed “gm,0” to “g0,m”.
- Figure 2, it’s logical to start from model 1 in the figure legend.
Response: In this study, there are too many models mentioned in the “Theoretical background”. We indicated “Leaf stomatal conductance (gsw, mol H2O·m-2·s-1) estimated by different models (Eqs. (1) and (2))”, so readers will clearly know what equation was used to calculate gsw.
- In contrast, when the effect of soil water stress was incorporated in the coupled Pn,L-gsw model (Eq. 2), the simulated gsw generally increased as SWC increased, regardless of the value imposed by qs. More specifically, at any given Ca, the amplification in simulated gs
Please, indicate what the qs is and how it is expressed. It is not given in equation (2).
Response: The qs is expressed in Page 4, line 122. Considering the reviewer’s suggestion, we have added a table of important notation to a supplementary material in page 26, line 687.
- Figure 3. Same as 15).
Response: Again, there are too many models mentioned in the “Theoretical background” in this study. We indicated “modeled leaf stomatal conductance (gsw, mol H2O·m-2·s-1) using different models (Eqs. (1) and (2))”, so readers will clearly know what equation was used to calculate gsw.
- Figure 4. Pls, indicate where the results di differ statistically. Same for all the figures.
Response: In this study, the influences of Ca and SWC on gsw, gm, and WUE (including WUEi,L, WUEi,P, and WUEs,P) were determined by two-way analysis of variance (ANOVA), and results were considered statistically significant at p < 0.05. However, the differences among treatments were not analyzed, as it is not very relevant to the main purpose of the manuscript.
- Figure 5. What are the parts a-f of the figure? Pls, indicate. The same for different values of qs.
Response: Considering the reviewer’s suggestion, we have described what the parts a-f of the figure in Page 15, line 380-383. Leaf mesophyll conductance (gm, mol CO2·m-2·s-1) were estimated by different models (Eqs. (3), left panels; and (4), right panels) under five soil water contents (SWC) × three CO2 concentrations (Ca). C400, C600 and C800 are Ca of 400 (a and b), 600 (c and d) and 800 µmol·mol-1 (e and f).
- Figures 6-10 are heavily overloaded with curves and data, some points are overlapping and not visible.
Response: Considering the reviewer’s suggestion, we have modified all figures and transformed some figures to tables.
- Figure 7, it seems better to divide and present separately the measured and modelled results.
Response: Considering the reviewer’s suggestion, we have divided and presented the measured and modelled results separately in Page 18, line 416.
- Basically it is not clear which models are considered and where.
Response: Considering the reviewer’s suggestion, we have made clear which models are considered.
gsw: the coupled Pn,L-gsw model, incorporating the water stress-dependent function with well parameterized qs (Eq. (2)), slightly more accurately captured the observed gsw pattern than the model excluding the soil water stress effect (Eq. (1)). –Page 12, line 346-349
gm: the proposed coupled Pn,L-gm model with well parameterized qm (Eq. (4)) effectively improved the predictive accuracy of gm compared to the previously introduced gm,p- and SWC-dependent model (Eq. (3)). –Page 16, line 392-394.
WUEi,L and WUEi,P: the C1 for WUEi,L and WUEi,P behaved slightly better than the C2. –Page 18, line 427-428
WUEs,P: the observed WUEs,P could be better represented by C1 than by C2. –Page 20, line 452-453
- The reasonable ideas and data are not presented well for a reader to understand clearly and potentially reproduce the results. It potentially (the Reviewer is not suggesting a single solution) might be reasonable to move some models to methodical part or to a supplementary material while transforming some figures to tables. So far, the submitted MS is not well presented and requires more information about the soil and illumination used for experiments.
Response: Considering the reviewer’s suggestion, we have transformed some figures to tables. To make it easier to understand, we have added a table of important notation to a supplementary material. In addition, we have rewritten the “Material and Methods”, more information about the soil and illumination used for experiments were presented.
# Reviewer 2:
- I am having trouble evaluating the model because I am not sure to what purpose it would be applied, and therefore how closely it should fit the data, especially with regard to CO2.
Response: The purpose of the study was to check the applicability of the whole-plant WUE model scaled from the leaf level, based on estimations of stomatal and mesophyll behavior. The C1 inferred from Eqs. (2) and (4) could more accurately represent the observed WUEs,P than the C2 inferred from Eqs. (1) and (3). The sensitivities of WUEs,P in responds to CO2 were not analyzed, as the developed model for estimating WUEs,P contains Ca (CO2 concentration) parameter. Instead, the uncertainties of WUEs,P associated with simulations of stomatal and mesophyll behavior were determined in Page 22-23, line 516-528.
- Some experimental details are missing - what was the light type and amount inside the chambers? Only two chambers were used, but there were 3 CO2 treatments - how was that accomplished? This is described in the title as "whole plant", but the root systems seems to have been excluded from any gas exchange measurements, or am I misinterpreting the methods?
Response: Considering the reviewer’s suggestion, we rewritten the “Material and Methods”, more information about the light type and amount inside the chambers were presented in Page 7, line 204-209. And we have described that how it experimentally possible to ensure 2 levels of CO2 (600 and 800 μmol·mol-1) in one chamber in Page 7, line 210-215. In this study, root systems have been excluded from gas exchange measurements, due to it was impossible to separate root and soil respiration for technical restriction (Page 23, line 541-543).
- I can not accept, in general, that gm responds to CO2 with the same sensitivity as gs, since there is a lot of data which contradicts that. In general, I do not accept the gs = m + A*H/C type equations, since they have gs going to m when C is low enough for A to = 0, which does not happen experimentally, although the equations may work well enough at high CO2.
Response: In this study, both gsw and gm decreased with elevated Ca, however, the amplification in gsw and gm by decreasing Ca at different soil water content is different. The sensitivities of gsw and gm in responds to CO2 were not analyzed, as it were not the focus of study. In addition, we acknowledge the shortcomings of our research. We only compare the measured results with simulations at ambient (400 μmol·mol−1) and elevated CO2 concentrations (600 and 800 μmol·mol−1). So the gs = m + A*H/C type equations (i.e. the coupled Pn,L-gsw model in the manuscript) worked well.
- Your data actually show very little, if any, response of either gs or gm to these CO2 treatments, which is a fairly common response in tree species. That is why your models struggle to fit that data. Mostly your gs and gm values are varying with soil water.
Response: In this study, changes in Ca led to significant effects on gsw and gm. Both gsw and gm decreased with elevated Ca, which was also reported previously (Wang et al., 2000; Robredo et al., 2010; Flexas et al., 2007), However, the amplification in gsw and gm by decreasing Ca at different soil water content is different.
- Flexas J, Diaz-Espejo A, Galmés J, Kaldenhoff R, Medrano H, Ribas-Carbo M. Rapid variations of mesophyll conductance in response to changes in CO2 concentration around leaves[J]. Plant Cell and Environment, 2007, 30(10): 1284-1298.
- Robredo A, Pérez-López U, Lacuesta M, Mena-Petite A, Muñoz-Rudea A (2010) Influence of water stress on photosynthetic characteristics in barley plants under ambient and elevated CO2 Biol Plant 54(2):285-292.
- Wang X, Curtis PS, Pregitzer KS, Zak DR (2000) Genotypic variation in physiological and growth responses of Populus tremuloides to elevated atmospheric CO2 Tree Physiol 20:1019-1028.

Reviewer 3 Report
I am having trouble evaluating the model because I am not sure to what purpose it would be applied, and therefore how closely it should fit the data, especially with regard to CO2.
Some experimental details are missing - what was the light type and amount inside the chambers? Only two chambers were used, but there were 3 CO2 treatments - how was that accomplished? This is described in the title as "whole plant", but the root systems seems to have been excluded from any gas exchange measurements, or am I misinterpreting the methods?
I can not accept, in general, that gm responds to CO2 with the same sensitivity as gs, since there is a lot of data which contradicts that. In general, I do not accept the gs = m + A*H/C type equations, since they have gs going to m when C is low enough for A to = 0, which does not happen experimentally, although the equations may work well enough at high CO2.
Your data actually show very little, if any, response of either gs or gm to these CO2 treatments, which is a fairly common response in tree species. That is why your models struggle to fit that data. Mostly your gs and gm values are varying with soil water.
Author Response
Dear Reviewer:
Thank you very much for your attention and the reviewers’ evaluation and comments on our paper “Scaling-up from leaf to whole-plant level for water use efficiency estimates based on stomatal and mesophyll behavior in Platycladus orientalis”. We have learned much from the comments which are valuable and helpful for revising and improving our manuscript, as well as the important guiding significance to our research. After carefully studying the comments and making corresponding correction, we have made corresponding changes. Revise portion are marked in red in the manuscript. The main corrections in the paper and responses to the reviewers’ comments are as following:
- I am having trouble evaluating the model because I am not sure to what purpose it would be applied, and therefore how closely it should fit the data, especially with regard to CO2.
Response: The purpose of the study was to check the applicability of the whole-plant WUE model scaled from the leaf level, based on estimations of stomatal and mesophyll behavior. The C1 inferred from Eqs. (2) and (4) could more accurately represent the observed WUEs,P than the C2 inferred from Eqs. (1) and (3). The sensitivities of WUEs,P in responds to CO2 were not analyzed, as the developed model for estimating WUEs,P contains Ca (CO2 concentration) parameter. Instead, the uncertainties of WUEs,P associated with simulations of stomatal and mesophyll behavior were determined in Page 22-23, line 516-528.
- Some experimental details are missing - what was the light type and amount inside the chambers? Only two chambers were used, but there were 3 CO2 treatments - how was that accomplished? This is described in the title as "whole plant", but the root systems seems to have been excluded from any gas exchange measurements, or am I misinterpreting the methods?
Response: Considering the reviewer’s suggestion, we rewritten the “Material and Methods”, more information about the light type and amount inside the chambers were presented in Page 7, line 204-209. And we have described that how it experimentally possible to ensure 2 levels of CO2 (600 and 800 μmol·mol-1) in one chamber in Page 7, line 210-215. In this study, root systems have been excluded from gas exchange measurements, due to it was impossible to separate root and soil respiration for technical restriction (Page 23, line 541-543).
- I can not accept, in general, that gm responds to CO2 with the same sensitivity as gs, since there is a lot of data which contradicts that. In general, I do not accept the gs = m + A*H/C type equations, since they have gs going to m when C is low enough for A to = 0, which does not happen experimentally, although the equations may work well enough at high CO2.
Response: In this study, both gsw and gm decreased with elevated Ca, however, the amplification in gsw and gm by decreasing Ca at different soil water content is different. The sensitivities of gsw and gm in responds to CO2 were not analyzed, as it were not the focus of study. In addition, we acknowledge the shortcomings of our research. We only compare the measured results with simulations at ambient (400 μmol·mol−1) and elevated CO2 concentrations (600 and 800 μmol·mol−1). So the gs = m + A*H/C type equations (i.e. the coupled Pn,L-gsw model in the manuscript) worked well.
- Your data actually show very little, if any, response of either gs or gm to these CO2 treatments, which is a fairly common response in tree species. That is why your models struggle to fit that data. Mostly your gs and gm values are varying with soil water.
Response: In this study, changes in Ca led to significant effects on gsw and gm. Both gsw and gm decreased with elevated Ca, which was also reported previously (Wang et al., 2000; Robredo et al., 2010; Flexas et al., 2007), However, the amplification in gsw and gm by decreasing Ca at different soil water content is different.
- Flexas J, Diaz-Espejo A, Galmés J, Kaldenhoff R, Medrano H, Ribas-Carbo M. Rapid variations of mesophyll conductance in response to changes in CO2 concentration around leaves[J]. Plant Cell and Environment, 2007, 30(10): 1284-1298.
- Robredo A, Pérez-López U, Lacuesta M, Mena-Petite A, Muñoz-Rudea A (2010) Influence of water stress on photosynthetic characteristics in barley plants under ambient and elevated CO2 Biol Plant 54(2):285-292.
- Wang X, Curtis PS, Pregitzer KS, Zak DR (2000) Genotypic variation in physiological and growth responses of Populus tremuloides to elevated atmospheric CO2 Tree Physiol 20:1019-1028.

Round 2
Reviewer 2 Report
The present Reviewer is definitely assured that the Authors present interesting results which under any circumstances could provide further discussion and progress in the area of research. The Authors basically replied to the posed questions with reasonable mostly comprehensive depth of arguments.
Still, however, a few questions remain which hamper the Reviewer to recommend the Manuscript for acceptance without revisions.
1) It is reasonable to add the concentrations of CO2 used (400-600-800) in the abstract.
2) The reference style of the journal requires references to be given as number according to the numerical order of their positions in the text of the Manuscript (not surnames of the Authors). Pls, correct accordingly.
3) Please, check the use of language again, especially for the new added parts.
For example, the one of the new added phrase requires substantial correction at least
“The collected values remained relatively stable from the 21th, 423 these were used for data analysis.”
4) It seems that were several experiments since one chamber was used for 600 μmol·mol−1 of CO2 in one series of experiments while for 800 μmol·mol−1 in the other series of experiments. Please, mention the fact and how the control treatment was chosen then (control for each series or common control etc.).
It looks like there were 15 plants at 600 μmol·mol−1 of CO2 and 15 plants at 800 μmol·mol−1 of CO2 (3X5 when the experiments lasted for 6 months with 5 plants per a month).
5) Please, indicate that LEDs provided “white” light, not e.g. red or blue LEDs only.
6) Still the method of leaf area measurements inspires questions. Were the leaves homogeneous then with similar width, shapes etc.
7) Figures 3 and 5 etc. are not corrected for the origin of q and so can not be understood. The figure legend for 3 mentions equation (2) (line 152 of the submitted MS), no q is given in the equation. Pls, explain all the introduced parameters either in the earlier equation or in the figure legends.
Same for figure 5.
What are the tunable parameters and their meaning?
8) Again. Figure 5. What are the parts a-f of the figure? Pls, indicate in the figure legend, not in the text. The same for different values of q which are still not explained earlier.
9) Figure legend 4, part of it. “water contents (SWC) Figure 400. C600 and C800 are Ca of 400 (a), 600 (b) and 800 μmol·mol-1 (c). Data represent mean values 615 ± SD. 616”
Where is the figure 400? Please, check the figure legends.
10) Still the text requires essential revision to be clear and understandable.
Author Response
Dear Reviewer:
Thank you very much for your attention and the reviewers’ evaluation and comments on our paper “Scaling-up from leaf to whole-plant level for water use efficiency estimates based on stomatal and mesophyll behaviour in Platycladus orientalis”. We have learned much from the comments which are valuable and helpful for revising and improving our manuscript, as well as the important guiding significance to our research. After carefully studying the comments and making corresponding correction, we have made corresponding changes. Revise portion are marked in red in the manuscript. The main corrections in the paper and responses to the reviewer’ comments are as following:
- It is reasonable to add the concentrations of CO2 used (400-600-800) in the abstract.
Response: Considering the reviewer’s suggestion, we have added the concentrations of CO2 used (400-600-800) in the abstract. In this study, an isotope model was scaled-up from the leaf to the whole-plant level, in order to simulate the variation of WUEs,P in response to different CO2 concentrations (Ca; 400, 600 and 800 μmol·mol−1) and soil water content (SWC; 35%-100% of field capacity) (Page 1, line 11-14).
- The reference style of the journal requires references to be given as number according to the numerical order of their positions in the text of the Manuscript (not surnames of the Authors). Pls, correct accordingly.
Response: Considering the reviewer’s suggestion, we have presented references as number according to the numerical order of their positions in the text of the Manuscript.
- Please, check the use of language again, especially for the new added parts.
For example, the one of the new added phrase requires substantial correction at least “The collected values remained relatively stable from the 21th, these were used for data analysis.”
Response: We are very sorry for our incorrect writing. We have changed the sentence as “The values used for data analysis are those that remained relatively stable from the 21st day of orthogonal treatments” (Page10, line 269-270). In addition, the use of language of the whole manuscript have been checked and corrected.
- It seems that were several experiments since one chamber was used for 600 μmol·mol−1 of CO2 in one series of experiments while for 800 μmol·mol−1 in the other series of experiments. Please, mention the fact and how the control treatment was chosen then (control for each series or common control etc.).
It looks like there were 15 plants at 600 μmol·mol−1 of CO2 and 15 plants at 800 μmol·mol−1 of CO2 (3X5 when the experiments lasted for 6 months with 5 plants per a month).
Response: Considering the reviewer’s suggestion, we have partly rewritten the “Introduction” and “Material and Methods”. The latest observations showed that globally averaged atmosphere CO2 concentration (Ca) reached new high (413.2 ± 0.2 µmol·mol-1) in 2020 (WMO Greenhouse Gas Bulletin 2021). If the upward trend of Ca continues, soil water stress may be intensified by climate change in many areas. Making it crucial to predict how WUEs,P responds to the different Ca and soil water content (SWC). (Page 3, line 78-81). That is why we chosen different Ca × SWC.
CO2 sensors and control system inside the chambers can continuously monitor and adjust Ca steady near the enactment value, with a standard deviation of 50 μmol•mol-1 (Page 8, line 228-229). That is how different Ca levels achieved.
As one growth chamber was able to hold five pots, the experiment was performed progressively from June to November 2018, where treatments were maintained at C400 × SWC ( in chamber b) and C600 × SWC (in chamber a) from June to August, and at C400 × SWC (in chamber b) from September to November. That is why the experiments lasted for 6 months with 5 plants per a month (Page 8, line 239-242).
- Please, indicate that LEDs provided “white” light, not e.g. red or blue LEDs only.
Response: Considering the reviewer’s suggestion, we have indicated the LEDs type. “From 07:00 to 19:00 (simulating daytime), all white LED lights were turned on... From 19:00 to 07:00 (simulating nighttime), all white LED lights were turned off…” (Page 7, line 216-218)
- Still the method of leaf area measurements inspires questions. Were the leaves homogeneous then with similar width, shapes etc.
Response: Considering the reviewer’s suggestion, we have added informations about leaves. A portion of leaves with different widths and shapes were selected as subsamples (Page11, line 310-311).
- Figures 3 and 5 etc. are not corrected for the origin of q and so can not be understood. The figure legend for 3 mentions equation (2) (line 152 of the submitted MS), no q is given in the equation. Pls, explain all the introduced parameters either in the earlier equation or in the figure legends.
Same for figure 5. What are the tunable parameters and their meaning?
Response: The equation (2) involved parameter of f(θs) which was inferred from qs (Eq. 5). Considering the reviewer’s suggestion, we have explained what qs was in the figure legends. qs is tunable parameter associated with stomatal limitation (Page 12, line 350).
- Figure 5. What are the parts a-f of the figure? Pls, indicate in the figure legend, not in the text. The same for different values of q which are still not explained earlier.
Response: Considering the reviewer’s suggestion, we have indicated what were the parts a-f of the figure. The estimated leaf mesophyll conductance (gm, mol CO2·m-2·s-1) is based on different models (Eq. (3), a, c and e; and Eq. (4), b, d and f) under varying soil water contents (SWC) and CO2 concentrations (Ca). C400, C600 and C800 are Ca of 400 (a and b), 600 (c and d) and 800 µmol·mol-1 (e and f). qm is tunable parameter associated with mesophyll limitation. Data represent mean values ± SD (Page 15, line 396-101).
- Figure legend 4, part of it. “water contents (SWC) Figure 400. C600 and C800 are Ca of 400 (a), 600 (b) and 800 μmol·mol-1 (c). Data represent mean values”
Where is the figure 400? Please, check the figure legends.
Response: We are very sorry for our incorrect writing. We have changed the figure legends as “Fig. 4 Response of measured leaf mesophyll conductance (gm, mol CO2·m-2·s-1) to three CO2 concentrations (Ca) ×five soil water contents (SWC) for Platycladus orientalis saplings. C400, C600 and C800 are Ca of 400, 600 and 800 µmol·mol-1. Data represent mean values ± SD” (Page 14, line 384-385).
- Still the text requires essential revision to be clear and understandable.
Response: Considering the reviewer’s suggestion, we have partly rewritten the manuscript.

Reviewer 3 Report
Surely the title should be "shoot" not "whole plant", since that is what was measured. I am not convinced that Ca actually affected gs - with n = 3, these SD look too big at each water level for significance, even CO2 affects on gm are marginal. No statistical analysis of CO2 effects alone is given for either gs or gm. It should be pointed out that Ca effects on gm (or even gs) are not universal across species. The type of LED lamps should be provided. How much did Ca change during the 3 minutes in the light and dark (approximately) inside the closed measurement chamber? It seems that Ca inside the "400" chamber was not controlled or measured - it could have been much lower in daytime and higher at night, depending on the rate of air exchange with outside, and the outside air CO2.
Author Response
Dear Reviewer:
Thank you very much for your attention and the reviewers’ evaluation and comments on our paper “Scaling-up from leaf to whole-plant level for water use efficiency estimates based on stomatal and mesophyll behaviour in Platycladus orientalis”. We have learned much from the comments which are valuable and helpful for revising and improving our manuscript, as well as the important guiding significance to our research. After carefully studying the comments and making corresponding correction, we have made corresponding changes. Revise portion are marked in red in the manuscript. The main corrections in the paper and responses to the reviewers’ comments are as following:
- Surely the title should be "shoot" not "whole plant", since that is what was measured.
Response: Considering the reviewer’s suggestion, we have indicated “measurements of whole-plant net CO2 gas exchange (root systems have been excluded from measurements, i.e., aboveground measurements) and transpiration under different Ca × SWC conditions were conducted” in the Introduction (page 3 line 58-86). The aboveground CO2 gas exchange measurements including whole leaves, twigs and stems. So the readers will clearly know what were measured in this study.
- I am not convinced that Ca actually affected gs - with n = 3, these SD look too big at each water level for significance, even CO2 affects on gm are marginal. No statistical analysis of CO2 effects alone is given for either gs or gm. It should be pointed out that Ca effects on gm (or even gs) are not universal across species.
Response: Considering the reviewer’s suggestion, we have pointed out that Ca effects on gm (or even gs) are not universal across species. That is “In addition, this study found that mostly gsw (and gm) values were varying with SWC, even if the influence of Ca on gsw (and gm) were significant. Centritto et al. (2002) also found that gsw was significantly lower in water-stressed seedlings than in well-watered seedlings, while elevated Ca didn’t significantly influence on gsw under either well-watered or water-stressed conditions. However, Flexas et al. (2002) observed that both gsw and gm were much higher in 400 µmol·mol-1 air than those in 1000 µmol·mol-1 air. Thus Ca effects on gsw (or even gm) could be not universal across species” (Page 21, line 506-511).
- The type of LED lamps should be provided.
Response: Considering the reviewer’s suggestion, we have indicated the LEDs type. “From 07:00 to 19:00 (simulating daytime), all white LED lights were turned on... From 19:00 to 07:00 (simulating nighttime), all white LED lights were turned off…” (Page 7, line 216-218).
- How much did Ca change during the 3 minutes in the light and dark (approximately) inside the closed measurement chamber?
Response: Considering the reviewer’s suggestion, we have indicated how did Ca change during the 3 minutes in the light and dark (approximately) inside the closed measurement chamber. During the 3 minutes (Page 9, line 258-259), the Ca in the closed chamber gradually decreased in daytime while increased in nighttime. As the result of difference in amplitude of variations among different treatments were obvious, how much did Ca change was not indicated.
- It seems that Ca inside the "400" chamber was not controlled or measured - it could have been much lower in daytime and higher at night, depending on the rate of air exchange with outside, and the outside air CO2.
Response: Considering the reviewer’s suggestion, we have indicated how Ca inside the "400" chamber was controlled and measured. That is “Another growth chamber (Fig. 1b) was only connected to ambient atmosphere with an intake pipe to maintain Ca of approximately 400 μmol·mol−1 (C400). CO2 sensors and control systems inside the growth chambers can continuously monitor and adjust Ca steady near the enactment value, with a standard deviation of 50 μmol·mol-1.” (Page 8, line 226-230). The Ca inside the "400" chamber remained relatively stable.

Round 3
Reviewer 2 Report
The Reviewer is basically satisfied by the answers of the Authors though still the Reviewer is mentioning a few minor changes which are preferably to be corrected.
1) Lines 30-31/
The water use efficiency[E1] (WUE), which refers to the ratio of carbon assimilation 31 to water transpired by plants (i.e., water loss), [E2]is essential in optimizing plant water 32 use[1].
What are the [E1] and [E2] which are not used later on?
2) The values used for data analysis are those that remained 282 relatively stable from the 21st [E3]day of orthogonal treatments.
What’s E3 here?
3) The θc was measured by soil water content (SWC) sensors (Onset. USA, 212 HOBO–U30) after soil samples were absorbed in water for 24 h with no vertical underwa- 213 ter droplets.
Pls, correct the phrase, it’s not correct. Absorbed water or were submerged in water or etc.
4) which was used to maintain elevated Ca of 600 μmol·mol−1 (C600) and 800 μmol·mol−1 234 (C800). Another growth chamber
Still not clear, better: 600 or 800, not 600 and 800.
5) Figure 1, figure legend. What are the and b mean? Pls, indicate in the figure legend.
6) The correlation between the measured and calculated gsw is shown in Table 1. When 370 applying Eq. (2), we found a strong correlation between the calculated and the 371 [Editor4]measured gsw (p < 0.01), and the correlation coefficient R2 decreased from 0.88 to 372 0.68 as qs increased from 0.25 to 1.50. The [E5]calculated gsw based on Eq. (1) also signifi- 373
Is it so important to mention respectable Editors in the text? The same is relevant to other parts of the text.
7) Figure legend qs is tunable parameter associated with stomatal limitation. The Reviewer is not able after looking at the Manuscript for the third time to see any tunable parameter in equation (2) given at line 144 of the MS, so leaves this point for the Editor and for the Readers if any. It’s finally the responsibility of the Authors to prove their texts and respond further on. The same is for this tunable parameter for figure 5 and tables 1 and 2, what the parameter is for? Which function it is added to? What is the physical meaning?
8) Figure 6. What are parts a-d of the figure? Pls, indicate in the figure legend.
9) Obviously, in model configuration 1 (Eq. (16)), gm was calculated by Eq. (4) with well 199 parameterized qm (qm = 0.25, see Results 3.2), and gsw was calculated by Eq. (2) with well 200 parameterized qs (qs = 0.25, see Results 3.1). In model configuration 2 (Eq. (17)), gm was 201 calculated by Eq. (3) with well parameterized qm (qm = 0.50, see Results 3.2), and gsw was 202 calculated by Eq. (1).
Part 3.1 is for methods. Same at line 149.
10) It’s impressive to see how the number of Authors changed.
Author Response
Dear Reviewer:
Thank you very much for your attention and the reviewers’ evaluation and comments on our paper “Scaling-up from leaf to whole-plant level for water use efficiency estimates based on stomatal and mesophyll behaviour in Platycladus orientalis”. We have learned much from the comments which are valuable and helpful for revising and improving our manuscript, as well as the important guiding significance to our research. After carefully studying the comments and making corresponding correction, we have made corresponding changes. Revise portion are marked in red in the manuscript. The main corrections in the paper and responses to the reviewers’ comments are as following:
- Lines 30-31: The water use efficiency[E1] (WUE), which refers to the ratio of carbon assimilation 31 to water transpired by plants (i.e., water loss), [E2]is essential in optimizing plant water 32 use[1]. What are the [E1] and [E2] which are not used later on?
Response: The water use efficiency[E1] is WUE, which is used in Paged 1 line 30, and Page 2 line 38 and 39. The E2 is also water use efficiency (i.e., WUE). The WUE can be studied at different spatial scales, such as WUEi-L, WUEi-P, and WUEs-P.
- The values used for data analysis are those that remained 282 relatively stable from the 21st [E3]day of orthogonal treatments. What’s E3 here?
Response: In this study, each treatment (Ca × SWC) lasted for 30 days. The Ep values remained relatively stable from the 21st day of orthogonal treatments, which were used for data analysis (Paged 9 line 259-262).
- The θc was measured by soil water content (SWC) sensors (Onset. USA, 212 HOBO–U30) after soil samples were absorbed in water for 24 h with no vertical underwater droplets. Pls, correct the phrase, it’s not correct. Absorbed water or were submerged in water or etc.
Response: Considering the reviewer’s suggestion, we have changed “were absorbed in water” to “absorbed water” (Paged 7 line 196).
- which was used to maintain elevated Ca of 600 μmol·mol−1 (C600) and 800 μmol·mol−1 (C800). Another growth chamber… Still not clear, better: 600 or 800, not 600 and 800.
Response: Considering the reviewer’s suggestion, we have changed “600 and 800” to “600 or 800” (Paged 7 line 215).
- Figure 1, figure legend. What are the and b mean? Pls, indicate in the figure legend.
Response: Considering the reviewer’s suggestion, we have indicated what are the a and b mean in the figure legend. That is, One growth chamber (a) was used to maintain elevated CO2 concentration of 600 μmol·mol−1 or 800 μmol·mol−1. Another growth chamber (b) was used to maintain CO2 concentration of 400 μmol·mol−1(Paged 8 line 232-234).
- The correlation between the measured and calculated gsw is shown in Table 1. When 370 applying Eq. (2), we found a strong correlation between the calculated and the 371 [Editor4]measured gsw (p < 0.01), and the correlation coefficient R2 decreased from 0.88 to 372 0.68 as qs increased from 0.25 to 1.50. The [E5]calculated gsw based on Eq. (1) also signifi- 373. Is it so important to mention respectable Editors in the text? The same is relevant to other parts of the text.
Response: We are so sorry for our inappropriate description. The manuscript was carefully revised by one of the authors (Guodong Jia), not by other editors. We have indicated author’s contribution in the manuscript.
- Figure legend qs is tunable parameter associated with stomatal limitation. The Reviewer is not able after looking at the Manuscript for the third time to see any tunable parameter in equation (2) given at line 144 of the MS, so leaves this point for the Editor and for the Readers if any. It’s finally the responsibility of the Authors to prove their texts and respond further on. The same is for this tunable parameter for figure 5 and tables 1 and 2, what the parameter is for? Which function it is added to? What is the physical meaning?
Response: Considering the reviewer’s suggestion, we have indicated that equation (2) incorporated a soil water stress-dependent function (f(θs), calculated by Eq. (5)) (Paged 3 line 102). The equation (5) incorporated parameter qj which is a measure of the nonlinearity of the effects of soil water stress on the limiting mechanisms (Page 4 line 125-126). In Figure 3, tunable parameter qs is a measure of the nonlinearity of the effects of soil water stress on the stomatal limiting mechanisms (Paged 12 line 348-350). In Figure 5, tunable parameter qm is a measure of the nonlinearity of the effects of soil water stress on the mesophyll limiting mechanisms (Paged 15 line 381-383).
- Figure 6. What are parts a-d of the figure? Pls, indicate in the figure legend.
Response: Considering the reviewer’s suggestion, we have indicated what are parts a-d of the figure. That is, measured (a) and simulated (b) leaf water use efficiency (WUEi-L, mmol·mol-1), as well as and measured (c) and simulated (d) whole-plant level instantaneous water use efficiency (WUEi-P, mmol·mol-1) in different model configurations…(Paged 17 line 411-413).
- Obviously, in model configuration 1 (Eq. (16)), gm was calculated by Eq. (4) with well 199 parameterized qm (qm = 0.25, see Results 3.2), and gsw was calculated by Eq. (2) with well 200 parameterized qs (qs = 0.25, see Results 3.1). In model configuration 2 (Eq. (17)), gm was 201 calculated by Eq. (3) with well parameterized qm (qm = 0.50, see Results 3.2), and gsw was 202 calculated by Eq. (1). Part 3.1 is for methods. Same at line 149.
Response: Considering the reviewer’s suggestion, we have moved some description to the “Material and methods” in Page 9-10 line 268-279. So the readers will clearly know measured and modelled WUEi,P ( and WUEs,P) were determined.
The description of “The whole-plant instantaneous water use efficiency (WUEi,P, mmol·mol-1) is the ratio of whole-plant net photosynthetic rate (Pn,p, µmol·h-1) to transpiration rate (Ep, mmol·h-1)” is still in the “Theoretical background”, in order to indicate what WUEi,P was.
- It’s impressive to see how the number of Authors changed.
Response: We have indicated all the author’s contribution in the “Author Contributions Statement”.
